# Nuclear export and translation of circular repeat-containing intronic RNA in C9ORF72-ALS/FTD

Shaopeng Wang[1,2,9], Malgorzata J. Latallo[3,4,9], Zhe Zhang[1,2], Bo Huang[1,5], Dmitriy G. Bobrovnikov[3], Daoyuan Dong[1,2], Nathan M. Livingston [3,4], Wilson Tjoeng[1,2], Lindsey R. Hayes[2,6], Jeffrey D. Rothstein [2,6], Lyle W. Ostrow [6], Bin Wu [3,4,7 ✉] & Shuying Sun[1,2,8 ✉]

*C9ORF72* hexanucleotide GGGGCC repeat expansion is the most common genetic cause of amyotrophic lateral sclerosis (ALS) and frontotemporal dementia (FTD). Repeat-containing RNA mediates toxicity through nuclear granules and dipeptide repeat (DPR) proteins produced by repeat-associated non-AUG translation. However, it remains unclear how the intron-localized repeats are exported and translated in the cytoplasm. We use single molecule imaging approach to examine the molecular identity and spatiotemporal dynamics of the repeat RNA. We demonstrate that the spliced intron with G-rich repeats is stabilized in a circular form due to defective lariat debranching. The spliced circular intron, instead of premRNA, serves as the translation template. The NXF1-NXT1 pathway plays an important role in the nuclear export of the circular intron and modulates toxic DPR production. This study reveals an uncharacterized disease-causing RNA species mediated by repeat expansion and demonstrates the importance of RNA spatial localization to understand disease etiology.

[1] Department of Pathology, Johns Hopkins University School of Medicine, Baltimore, MD, USA. [2] Brain Science Institute, Johns Hopkins University School of Medicine, Baltimore, MD, USA. [3] Department of Biophysics and Biophysical Chemistry, Johns Hopkins University School of Medicine, Baltimore, MD, USA. [4] Center for Cell Dynamics, Johns Hopkins University School of Medicine, Baltimore, MD, USA. [5] Sol Goldman Pancreatic Cancer Research Center, Johns Hopkins University School of Medicine, Baltimore, MD, USA. [6] Department of Neurology, Johns Hopkins University School of Medicine, Baltimore, MD, USA. [7] Solomon H. Snyder Department of Neuroscience, Johns Hopkins University School of Medicine, Baltimore, MD, USA. [8] Department of Physiology, Johns Hopkins University School of Medicine, Baltimore, MD, USA. [9] These authors contributed equally: Shaopeng Wang, Malgorzata J. Latallo. ✉email: bwu20@jhmi.edu; shuying.sun@jhmi.edu

Expansions of short nucleotide sequence repeats account for more than 40 neurological or neuromuscular diseases[1]. The mechanisms of disease pathogenesis depend on the repeat sequence, the expansion size, the gene context, and the location. In particular, the diversity of repeat expansion sequence and length is most prominent in introns. Compared to the repeats located in the mRNA, the intronic expansions likely involve more complicated RNA processing pathways. However, most studies focused on pure repeats and rarely considered the influence of gene context and spatiotemporal localization of RNA molecules in vivo.

Hexanucleotide (GGGGCC) repeat expansion (HRE) in intron 1 of the *C9ORF72* gene is the most common genetic cause of both amyotrophic lateral sclerosis (ALS) and frontotemporal dementia (FTD)[2,3]. RNA-mediated gain of toxicity is implicated in the pathogenicity, including the formation of nuclear RNA granules which likely sequester RNA binding proteins and disrupt RNA processing[4,5], and production of toxic dipeptide repeat (DPR) proteins by repeat-associated non-AUG (RAN) translation[6–12]. However, the molecular identity and the spatial distribution of these repeat-containing RNAs remain elusive. The GGGGCC$^{exp}$ is located inside the first intron of the *C9ORF72* gene. Intron containing species, namely unspliced pre-mRNA and spliced intron, are normally degraded rapidly within the nucleus, and excluded from cytoplasm. It is an important aspect of ALS/FTD etiology to determine which RNA species are exported and subjected to RAN translation, and which are retained in the nucleus and form granules. Traditional biochemical or genetic approaches require breaking up cells and the spatiotemporal information is lost. We applied single-molecule imaging approaches to visualize the *C9ORF72* repetitive RNAs directly in cells to probe the molecular and biophysical properties of RNA granules and DPR biogenesis.

## Results

### Reporters for single-molecule visualization of *C9ORF72* RNAs.
To investigate the spatiotemporal dynamics of the repeat RNA in an intronic context, we designed a reporter containing 70× (GGGGCC) repeats in the first intron of *C9ORF72* flanked by the first two exons with the native exon–intron junction elements (Fig. 1a). To visualize RNA in live cells, we employed the MS2 system, in which 24× MS2-binding sites (MBS) inserted into the target RNA were fluorescently labeled by MS2 coat proteins (MCP) expressed in the same cell[13]. To simultaneously visualize both introns and exons, we utilized orthogonal PP7 binding sites (PBS) and PP7 coat proteins (PCP)[14]. We inserted 24×MBS[15] after the GGGGCC$^{exp}$ within the intron, and 24×PBS in the second exon (Fig. 1a). An identical construct without the HRE was used as a negative control. We stably expressed the two constructs separately in the U-2 Osteosarcoma (U-2 OS) cell line expressing stdMCP-Halotag and stdPCP-stdGFP[15]. The transcripts were efficiently spliced as expected, confirmed by RT-PCR (Supplementary Fig. 1a, b). We also performed nanopore long read sequencing of poly(A) selected mRNAs from the (GGGGCC)$_{70}$ reporter stable cells as well as transiently transfected HEK293T cells. Besides the expected 5′ truncated transcripts (read in the 3′→5′ direction), all the full-length reads containing PBS showed correct splicing to exon 1 (Supplementary Figs. 2 and 3). This indicates that the reporter can properly represent the endogenous *C9ORF72* splicing and is suitable for direct visualization of different RNA species: the unspliced pre-mRNA containing both MBS and PBS; the spliced intron with only MBS; and mature mRNA with only PBS (Fig. 1a).

### Repeat RNAs form granules in the nucleus.
We performed single-molecule Fluorescence In Situ Hybridization (smFISH)[16,17] to quantify the spatial distribution of various RNA species in different cellular compartments. We used separate probe sets labeled with distinct colors to target MBS (magenta, intron) and PBS (cyan, exon), respectively. In nucleus, we observed single RNAs as well as large RNA granules (Fig. 1b), as reported previously[18]. The small RNA clusters in no-repeat control cells likely represent transcription sites, as they were virtually abolished by transcription inhibition (Actinomycin D) (Supplementary Fig. 4a). The (GGGGCC)$_{70}$ cells showed more and larger granules, mostly contain both intron and exon signals (Fig. 1b, c). We quantified the numbers of introns and exons in each granule by normalizing the integrated fluorescence intensity to the corresponding single RNAs. The colocalized granules contained twice as many introns as exons, therefore likely to be composed of both spliced introns and unspliced pre-mRNAs; while in the RNA clusters in the control cells, the intron and exon numbers were similar (Fig. 1d). The pure-intron granules existed only in (GGGGCC)$_{70}$ cells (Fig. 1c, Supplementary Fig. 4b), smaller than exon–intron colocalized ones (Supplementary Fig. 4c). In addition, there were also more single spliced introns in (GGGGCC)$_{70}$ nuclei compared with controls (Fig. 1e). This was further confirmed by qRT-PCR that although the transgene expression level was lower in (GGGGCC)$_{70}$ cells than in controls (Supplementary Fig. 4d), the relative intron RNA level was significantly higher (Supplementary Fig. 4e). Altogether, this strongly suggests that the spliced *C9ORF72* intron is stabilized by the GGGGCC repeats.

### Spliced introns are exported into cytoplasm.
We next examined whether unspliced pre-mRNAs or spliced introns, or both are exported into cytoplasm. First, intronic RNAs were detected in the cytoplasm of (GGGGCC)$_{70}$ cells but not in controls (Fig. 1b, f). Over 95% of cells expressing the HRE construct had intron signals in cytoplasm, even though there were fewer cytoplasmic exons (mature mRNAs) in (GGGGCC)$_{70}$ cells than control (Fig. 1g), consistent with the lower transgene expression level (Supplementary Fig. 4d). Second, of the 6108 total counted cytoplasmic introns, over 98% did not colocalize with exon signals (Fig. 1h), indicating they were spliced introns, instead of unspliced pre-mRNAs. On the contrary, when mutating the 5' splice site (5'ss) of the reporter, majority of the cytoplasmic RNAs contain both MBS and PBS (Supplementary Fig. 4f). This further proved that splicing is required to generate the cytoplasmic MBS-only RNA (excised intron) in the splicing reporter.

We also performed live cell imaging to measure the motility of cytoplasmic RNA species. The exonic mRNA was tagged by stdPCP-stdGFP[15], while intron by stdMCP-Halotag labeled with JF646 Halo dyes[19,20]. We tracked single RNAs (Supplementary Movie 1) and observed similar average diffusion coefficients of spliced introns (0.35 μm$^2$/s) and mRNAs (0.25 μm$^2$/s) (Supplementary Fig. 4g, h)[21]. The slight difference is probably due to higher occupancy of ribosomes on mRNAs than introns due to more efficient canonical translation initiation[22–24]. Taken together, these findings suggest that the *C9ORF72* GGGGCC$^{exp}$ enables the spliced intron to be exported to cytoplasm, with comparable motility to mature mRNAs.

### Intron export in patient-derived cells.
We confirmed our findings with endogenous *C9ORF72* RNA in patient cells. We designed two sets of smFISH probes labeled with different fluorescent dyes, targeting endogenous *C9ORF72* intron 1 and all exons, respectively (Fig. 2a). To validate the specificity of the probe sets, we knocked down the *C9ORF72* gene in U-2 OS cells using the antisense oligonucleotide (ASO)[25] and examined

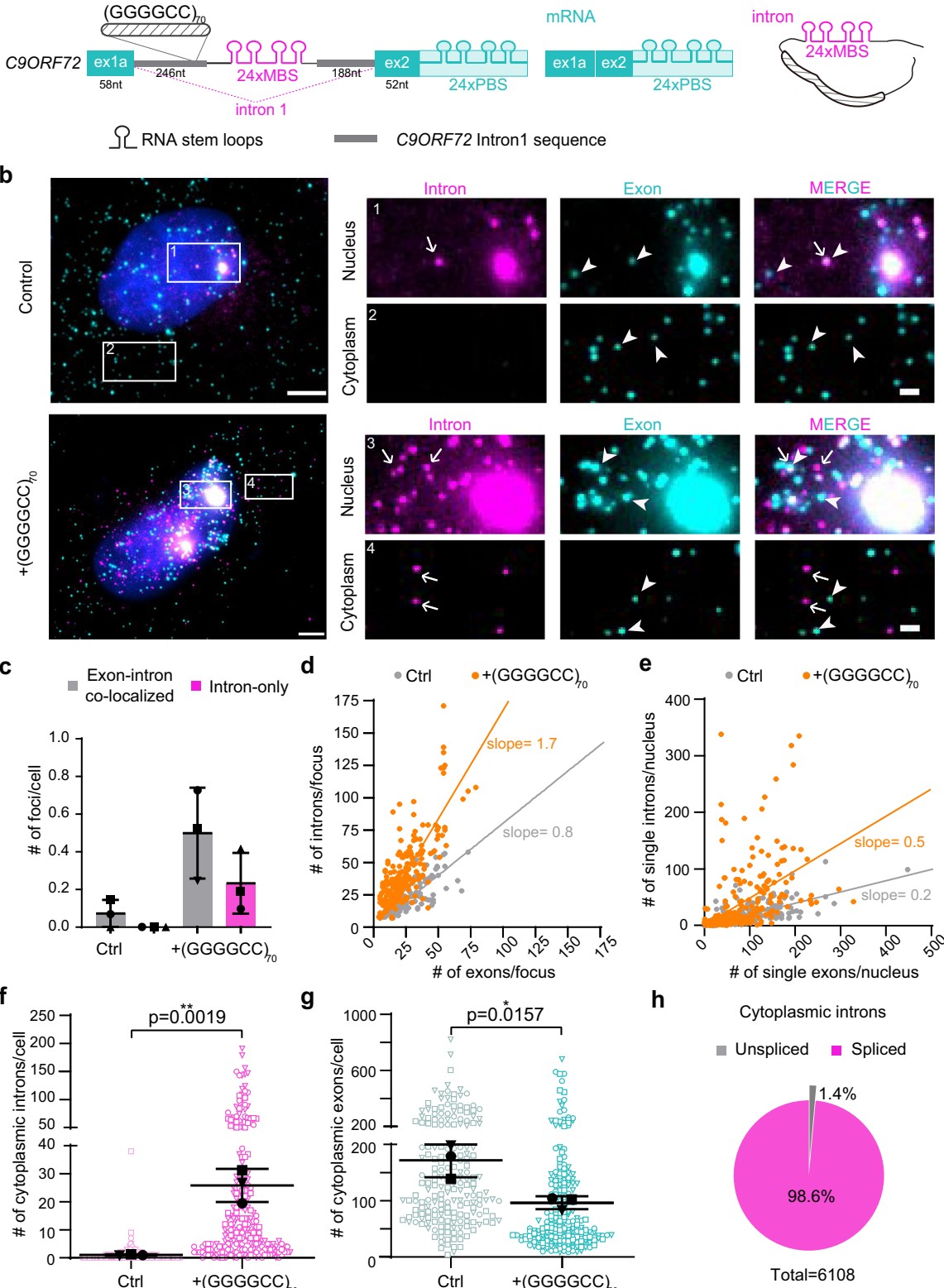

**a** C9ORF72 splicing reporter

**b**

**c** Exon-intron co-localized ▪ Intron-only

**d** ● Ctrl ● +(GGGGCC)$_{70}$

**e** ● Ctrl ● +(GGGGCC)$_{70}$

**f**

**g**

**h** Cytoplasmic introns

whether the RNA reduction can be recapitulated by smFISH. Compared to control cells, both the intron and exon RNA numbers per cell decreased in C9-ASO transfected cells (Supplementary Fig. 5a, b), and the reduction level was consistent with the qRT-PCR result (Supplementary Fig. 5c), demonstrating the specificity of the smFISH signals. We next used this method to

examine the endogenous *C9ORF72* RNA expression in patient cells. In immortalized lymphoblast cells, the nuclear RNA granules containing both introns and exons were readily observed in C9ORF72-ALS patient-derived cells, but not in healthy controls (Supplementary Fig. 5d). In order to better examine the RNA localization in cytoplasm, we performed smFISH on human

**Fig. 1 *C9ORF72* repeat-containing introns form nuclear RNA granules and are exported into cytoplasm. a** Schematic of the C9ORF72 splicing reporter construct. MBS/PBS: MS2/PP7 binding sites; ex1a, ex2: exon 1a and 2 of *C9ORF72* gene. Two sets of RNA FISH probes were designed to target either 24×MBS or 24×PBS with different fluorescent dyes. The mature mRNA only shows PBS signal, spliced intron only has MBS signal, and the unspliced pre-mRNA contains both MBS and PBS signals. **b** Representative images of two-color smFISH experiment to study the spatial distribution of RNA species in control or +(GGGGCC)$_{70}$ reporter cells. The boxes 1–4 were enlarged on the right. Magenta: intron (MBS); cyan: exon (PBS); blue: DAPI; arrow: single-intron molecule; arrowhead: single exon molecule. Scale bar: 5 μm and 1 μm for zoom in, respectively. Quantification shown in panels **c–g**. **c** Number of RNA granules per nucleus. Repeat-containing introns formed large granules colocalized with exons (gray) as well as intron only (pink) granules. Cells were treated with transcription inhibitor actinomycin D (1 μg/mL for 20 min) to exclude the effect of transcription. Data are mean ± SD from three biological replicates. The quantified cell number: Ctrl (88, 82, 41) and (GGGGCC)$_{70}$ (72, 43, 73). **d** Scatter plot of intron *vs* exon numbers in each RNA granule. Each dot represented one granule (from three biological replicates). The lines were linear fit to the scatter plot. The slope reflects the ratio of intron *vs* exon molecules in each granule. **e** Scatter plot of the numbers of single introns vs exons in each nucleus. Each dot represented a single nucleus. The lines were linear fit to the scatter plot. The slope reflects the ratio of intron *vs* exon molecules in each nucleus. **f, g** Quantification of cytoplasmic intron (**f**) and exon (**g**) number per cell. Each symbol represented a single cell and the three shapes represented three biological replicates. The mean of each replicate (larger black shapes) was used to calculate the average (horizontal bar) and SD (error bars) in each group, as well as for statistic comparison between groups. *$P < 0.05$, **$P < 0.01$, two-tailed *t*-test. The cell number in **d–g**: Ctrl (60, 43, 93) and (GGGGCC)$_{70}$ (93, 95, 49). **h** Percentage of cytoplasmic introns colocalized (unspliced) or uncolocalized (spliced) with exons in +(GGGGCC)$_{70}$ cells. The exported GGGGCC repeat-containing RNAs are predominantly spliced introns. Source data are provided as a Source Data File.

fibroblast cultures from skin biopsies (Fig. 2b). The nuclear granules were less evident in this cell type as the *C9ORF72* gene expression level is lower. The transfection of C9-ASO knocked down both the intron and exon RNA similar to the U-2 OS cells, as measured by qRT-PCR and smFISH (Supplementary Fig. 5e–g), confirming the specificity of the smFISH in patient cells. The levels of single mRNA molecules (exons) were similar in control and patient cells in both nucleus (Fig. 2c) and cytoplasm (Fig. 2d). Unspliced pre-mRNAs were slightly increased in C9ORF72-ALS cells (Fig. 2c), consistent with previous reports[26], but they were exclusively localized in nucleus. Spliced introns were detected in the cytoplasm of all three lines of C9ORF72-ALS patient cells, but were rarely found in healthy controls (Fig. 2e). And the cytosolic introns did not colocalize with exons (Fig. 2f), consistent with the reporter data (Fig. 1f, h). This demonstrated that the GGGGCC$^{exp}$ stabilizes the spliced intron of endogenous *C9ORF72* and mediates its export to cytoplasm in patient cells.

**The cytoplasmic introns are in circular form**. During splicing, the excised 5′ end of intron links to the branch site (usually "A" near the 3′ end of intron) to form a looped structure, known as a lariat, followed by 3′ splice site cleavage and exon–exon joining[27] (Fig. 3a). Most introns are subsequently debranched and quickly degraded after excision from nascent transcripts. However, stable circular introns, derived from inefficient lariat debranching, have been identified in mammalian cells, either in the nuclear insoluble fractions[28] or in the cytoplasm[29]. We therefore examined whether the cytoplasmic *C9ORF72* GGGGCC$^{exp}$-containing intron escaped debranching and stayed in a circular form. The exonuclease RNase R degrades only linear RNAs, but not circular ones (Fig. 3a). We developed a smFISH protocol that kept RNase R activity in fixed cells (Methods). We treated cells with RNase R or just buffer right before smFISH and quantified the RNA distribution and changes. In the experiments using various linear RNA reporter cells, RNase R treatment dramatically decreased mCherry-24×MBS mRNAs (Supplementary Fig. 6a, b), (GGGGCC)$_{70}$-24×MBS mRNAs (Supplementary Fig. 6c–e), and endogenous *POLR2A* mRNAs (Supplementary Fig. 7a–c). This demonstrates that the linear RNAs were efficiently degraded, and neither MBS/MCP nor (GGGGCC)$_{70}$ repeat structures inhibited the RNase R exonuclease activity. In the (GGGGCC)$_{70}$ splicing reporter cells, all the exons and nuclear introns were significantly reduced (Fig. 3b, d, Supplementary Fig. 7a, c), indicating that majority of them are linear transcripts (either pre-mRNAs or spliced and debranched introns in nucleus). Surprisingly, cytoplasmic introns persist even when treated with RNase R

(Fig. 3b, c). Therefore, the resistance of cytosolic introns to RNase R digestion demonstrates its circular form.

To pinpoint the molecular identity of the circular intronic RNA, we use RT-PCR with primers cross the potential branch site junction (Fig. 3e). The amplified fragment with the expected size was enriched in the RNase R-treated samples (Fig. 3f). We deep sequenced the PCR fragment and identified the junction of the 5′ splice site and the branch site (Fig. 3e). A novel upstream alternative 5′ splice site was mostly found to be connected with the branch site in the circular intronic RNA (Fig. 3e). The same 5′ splice site was also predominantly used in the mRNA isoforms (Supplementary Figs. 2, 3 and 7d). The dominant branch site is the "A" 40-nucleotide before the 3′ splice site (Fig. 3e). We also confirmed the circularity of endogenous *C9ORF72* intron in patient lymphoblast cells and postmortem motor cortex. The endogenous branch site junctions (Supplementary Fig. 8a) and 5′ splice sites (Supplementary Fig. 8b) were consistent with the ones identified in the reporter cells (Fig. 3e, Supplementary Fig. 7d), with slightly higher heterogeneity of the branch site usage. Altogether, this supports that the circular repeat-containing intron is derived from a stable lariat structure that escapes debranching.

**G-rich repeats stabilize the spliced intron and mediate the nuclear export**. Among the repeat expansion-linked diseases, the sequence diversity is more prominent in noncoding regions, particularly introns, than coding regions[30]. We therefore examined whether other expanded repeats can also induce the spliced intron stabilization and export. We chose myotonic dystrophy type 2 (DM2)-associated CCTG$^{exp}$ in *ZNF9* and spinocerebellar ataxia type 10 (SCA10)-associated ATTCT$^{exp}$ in *ATXN10* as representatives for intronic GC-rich and AT-rich repeats, respectively. We also tested the *C9ORF72* antisense CCCCGG$^{exp}$ and *FMR1* CGG$^{exp}$ in fragile X-associated tremor/ataxia syndrome (FXTAS). These two expansions were not originally found in introns but used in the C9-splicing context here as C-rich and G-rich repeat elements, respectively. We replaced the (GGGGCC)$_{70}$ in the C9ORF72 splicing reporter with (CCTG)$_{240}$, (ATTCT)$_{73}$, (CCCCGG)$_{70}$, or (CGG)$_{98}$ and stably expressed them in separate U-2 OS cell lines (Fig. 4a).

smFISH showed that although mRNAs (exons) were widely present in both nucleus and cytoplasm for all the reporter cells, significant amount of cytosolic introns was only found in the (CGG)$_{98}$ cells (Fig. 4b–e, Supplementary Fig. 9a, b). Similar to the (GGGGCC)$_{70}$ cells, the (CGG)$_{98}$ introns in the cytoplasm were predominantly spliced, not colocalized with exons (Fig. 4b). Neither the AT-rich repeats (ATTCT)$_{73}$, nor the C-rich repeats

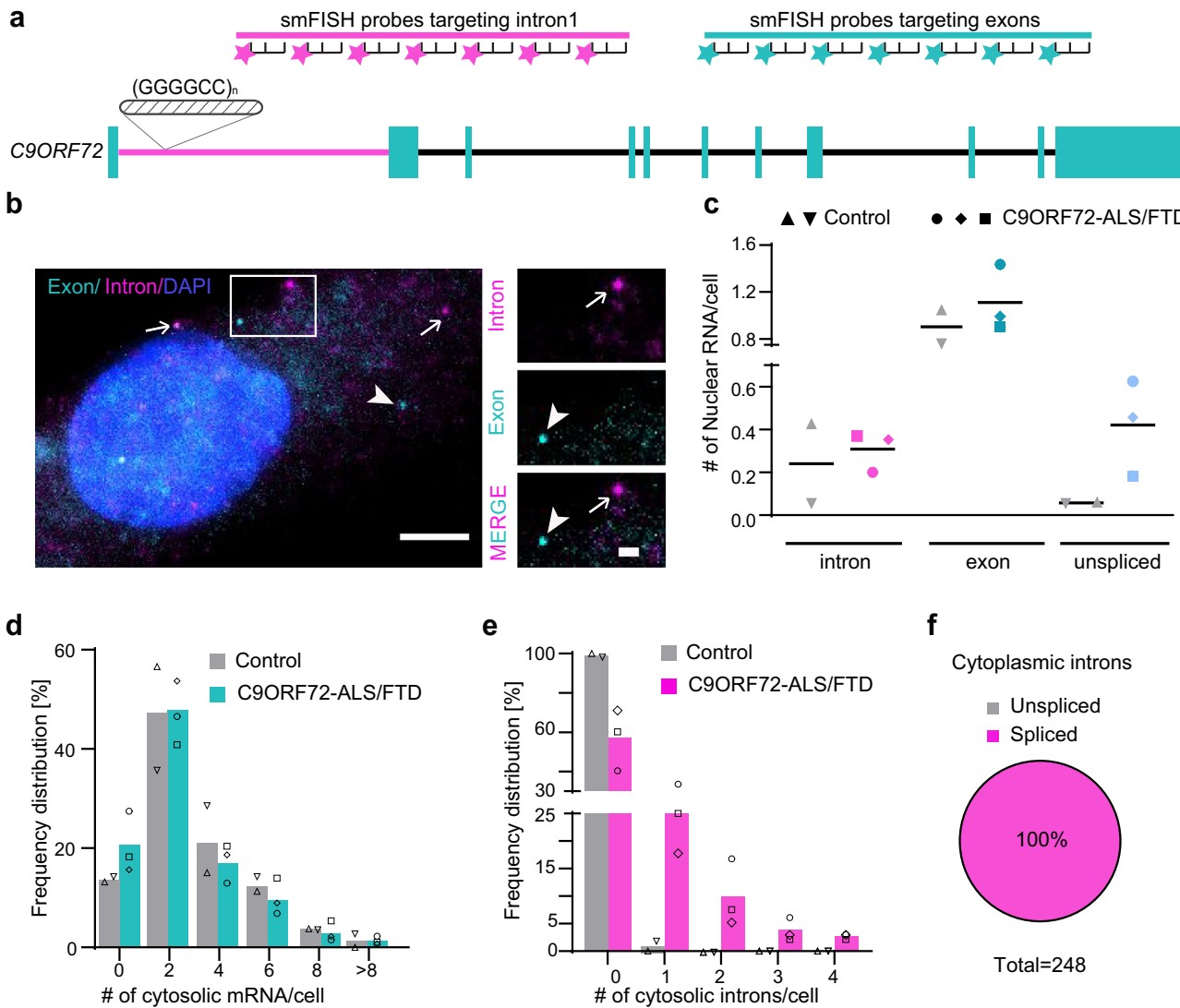

**Fig. 2 The distribution and molecular identity of endogenous *C9ORF72* RNAs in patient fibroblast cells. a** Schematic of smFISH probes targeting endogenous *C9ORF72* RNAs. A set of 76 probes targets intron 1 (magenta) and a 77-probe set targets all the exons (cyan). **b** Representative two-color smFISH images of endogenous *C9ORF72* RNAs in C9ORF72-ALS patient-derived fibroblasts. arrow: single-intron molecule; arrowhead: single exon molecule. Scale bar: 5 μm, and 1 μm for zoom in. **c** Quantification of nuclear introns, mRNAs and pre-mRNAs per nucleus. Datapoints of different shapes represent individual fibroblast line derived from two healthy controls and three C9ORF72-ALS patients. The horizontal bar indicated the mean value. **d**, **e** The percentage of cells containing different numbers of cytosolic exons (**d**) and introns (**e**) were quantified and calculated from each line with three technical replicates. Datapoints of different shapes represent individual fibroblast line derived from two healthy controls and three C9ORF72-ALS patients. The quantified cell number: Ctrl-1 (81), Ctrl-2 (165), C9-2 (135), C9-5 (131), and C9-6 (93). **f** Cytoplasmic introns were not colocalized with exons in C9ORF72-ALS patient fibroblasts, supporting the nuclear export of spliced intron. Source data are provided as a Source Data File.

$(CCTG)_{240}$ and $(CCCCGG)_{70}$, were able to induce the intron to be exported to the cytoplasm (Fig. 4c–e, Supplementary Fig. 9a, b). We also engineered the $(CCTG)_{240}$ and $(ATTCT)_{73}$ in their original gene context by constructing ZNF9- and ATXN10-splicing reporters, which showed no significant cytosolic spliced introns either (Supplementary Fig. 9c, d). For nuclear RNAs, the $(CGG)_{98}$ cells contained dramatically higher amount of single spliced introns than the other three repeats (Supplementary Fig. 10a). Only the $(CGG)_{98}$ granules contain higher number of introns than exons (Supplementary Fig. 10b), supporting the composition of both spliced introns and unspliced pre-mRNAs, similar to the $(GGGGCC)_{70}$ granules (Fig. 1d). Altogether, these results suggest that the G-rich sequences and/or secondary structures of expanded repeats are important to stabilize the spliced intron and mediate the export to cytoplasm.

**Translation of circular repeat-containing intron**. To analyze whether the spliced intron undergoes RAN translation in the cytoplasm, we utilized the single-molecule imaging of nascent peptides (SINAPS) technology[21],[31–33]. We placed 24× SunTag epitopes[34] between the $(GGGGCC)_{70}$ and MBS in the splicing reporter (Fig. 5a). As the RNA is translated, the nascent protein is immediately recognized by a fluorescent nanobody expressed in the same cell (single-chain variable fragment against SunTag peptide fused with super folder GFP (scFv-sfGFP)[21]. We transiently expressed the reporter in U-2 OS cells with membrane tethered stdMCP for long-term tracking of translating RNA molecules. The translation sites (TLS) detected by SunTag signals colocalized with single-intron RNA molecules (Supplementary Movie 2). When cells were treated with the translation inhibitor puromycin, the TLS disappeared while intron RNAs remained intact (Fig. 5b), which verified the active translation signals.

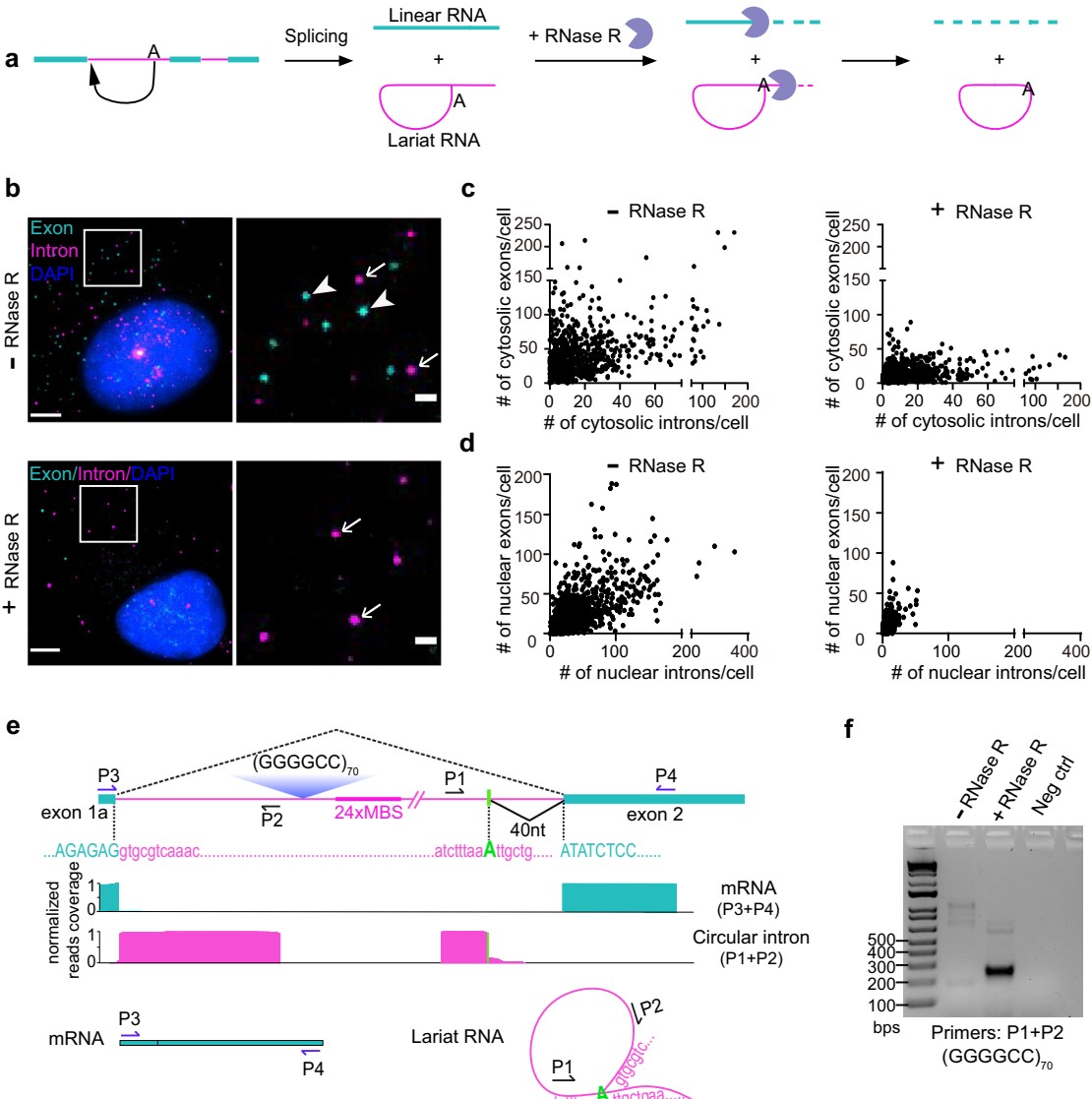

**Fig. 3 The cytoplasmic repeat-containing introns are exported in the circular form. a** Schematic of splicing and circular RNA enrichment by RNase R treatment. RNase R only degrades linear RNA. The tail of the lariat can be degraded, but the circular part will be preserved. **b** Representative images of two-color smFISH in +(GGGGCC)$_{70}$ reporter cells treated with RNase R (bottom) or just buffer (top). Magenta: intron (MBS); cyan: exon (PBS); blue: DAPI; arrow: intron; arrowhead: exon. Scale bar: 5 μm, and 1 μm for zoom in. Quantification shown in panels **c** and **d**. **c**, **d** Scatter plot of intron vs exon numbers in the cytoplasm (**c**) or nucleus (**d**) of each cell without RNase R (left) or with RNase R treatment (right). Each dot represented one cell. Total three technical replicates were carried out: 287, 289, 263 cells quantified for RNase R treatment and 222, 369, 342 cells for control (without RNase R treatment). **e** Deep sequencing identified circular boundaries of intron RNAs. The major splicing junction in mRNAs and branch site in circular RNAs were annotated. The libraries were prepared using the indicated primers (P1–P2 for circular intron; P3–P4 for mRNA), and the whole fragments were sequenced by 1 × 300nt MiSeq. The reads spanning the exon–exon junctions in mRNAs were aligned, shown in cyan. The reads spanning the junction between branch site and 5′-end of the intron in circular RNAs were aligned and shown in magenta. **f** RT-PCR of circular intron across the branch site boundary from +(GGGGCC)$_{70}$ reporter cells with or without RNase R treatment. The same amount of RNAs before and after RNase R treatment were used to synthesize the cDNA. PCR products using the divergent primers P1+P2 were only evident in RNase R-treated samples. Total three independent biological replicates were examined with similar results. Source data are provided as a Source Data File.

We created a stable cell line expressing the translation reporter in U-2 OS cells. By smFISH coupled with GFP immunofluorescence (IF), we identified 120 TLS out of 4768 cytoplasmic intron molecules, with no colocalization with exons (Fig. 5c, d). Similar fraction of translating introns remained after RNase R treatment, while exons were depleted (Supplementary Fig. 11a), confirming that the circular intron is the substrate of RAN translation. The fraction of translating introns (3%) and the number of nascent peptides (~2) per TLS is much lower than canonical AUG-dependent translation (50–90% RNA and 5–10 peptides per TLS)[21],

probably due to the inefficient cap-independent translation initiation of circular RNA. To further confirm that RAN translation of GGGGCC repeats can happen independent of 5′-cap, we performed in vitro translation assay using the in vitro transcribed bicistronic reporter RNAs. The first open reading frame (ORF) at the 5′ end is translated by the canonical cap- and AUG-dependent translation and terminated by multiple stop codons. The second cistron, Nanoluc luciferase (NLuc), did not have start codon ATG and was fused downstream of the (GGGGCC)$_{70}$ to monitor RAN translation (Supplementary Fig. 11b). We inserted either

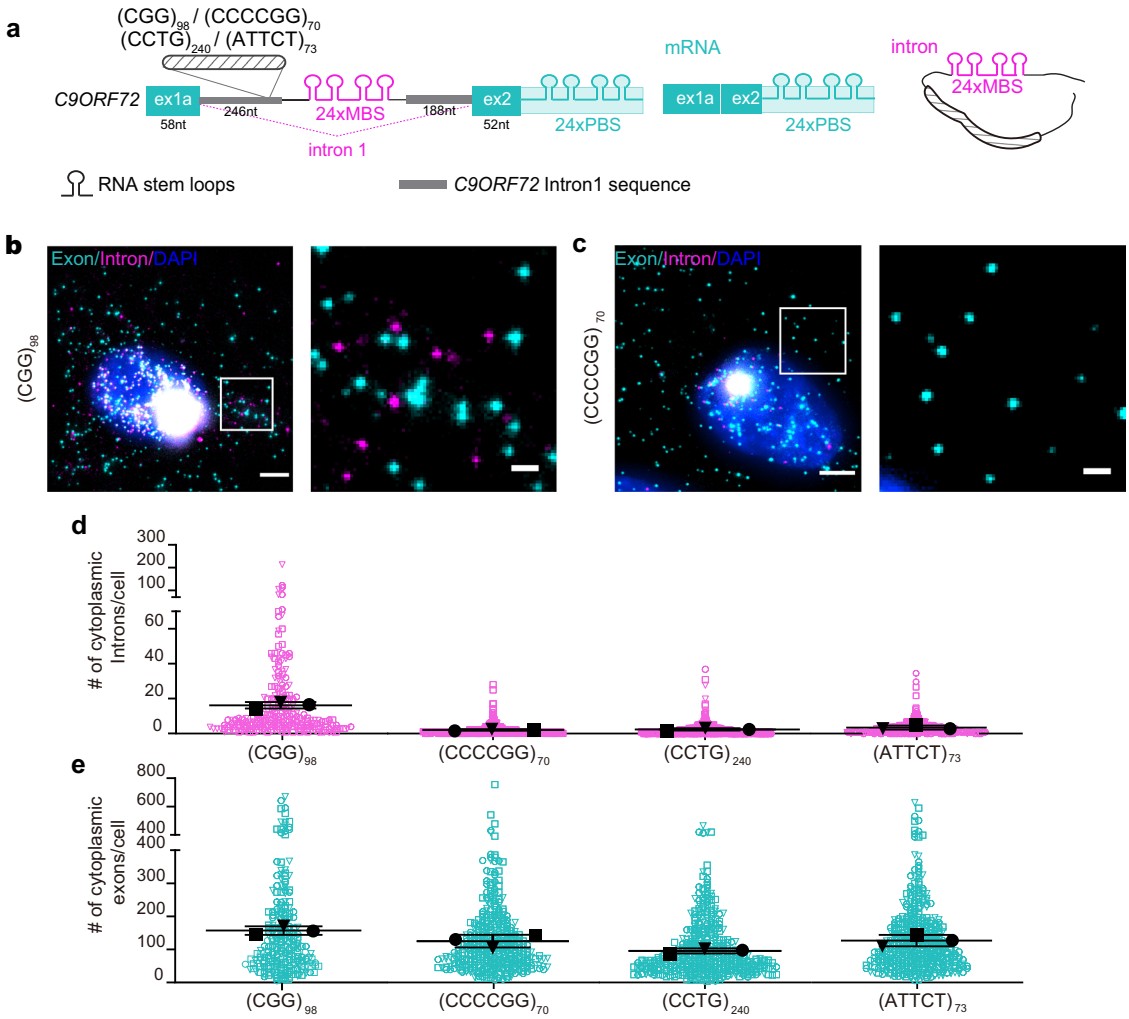

**Fig. 4 G-rich repeats stabilize the spliced intron and mediate the nuclear export. a** Schematic of splicing reporter constructs with four other repeat sequences in the context of *C9ORF72* gene. The GGGGCC$^{exp}$ was replaced with the indicated repeats in the reporter of Fig. 1a. MBS/PBS: MS2/PP7 binding sites; ex1a, ex2: exon 1a and 2 of *C9ORF72* gene. From these reporters, the mature mRNA only shows PBS signal, spliced intron only has MBS signal, and the unspliced pre-mRNA contains both MBS and PBS signals. **b**, **c** Representative images of two-color smFISH of C9ORF72 splicing reporter cells containing intronic (CGG)$_{98}$ (**b**) and (CCCCGG)$_{70}$ (**c**) repeats. Scale bar: 5 μm, and 1 μm for zoom in. **d**, **e** Quantification of cytoplasmic intron (**d**) and exon (**e**) numbers per cell in the splicing reporter cell lines with four different repeat expansions. Each symbol represented a single cell and the three shapes represented three biological replicates (*n* = 75, 80, 73 cells in (CGG)$_{98}$; *n* = 162, 146, 103 cells in (CCCCGG)$_{70}$; *n* = 147, 171, 119 cells in (CCTG)$_{240}$; *n* = 180,114,151 cells in (ATTCT)$_{73}$). The mean of each replicate (larger black shapes) was used to calculate the average (horizontal bar) and SD (error bars) in each reporter line. Source data are provided as a Source Data File.

randomized sequences before NLuc as negative control, or put independent ribosome entry site (IRES) sequences from either the encephalomyocarditis virus (EMCV) or cricket paralysis virus (CrPV) as positive controls (Supplementary Fig. 11b). The NLuc in both GA and GP frames showed comparable activities as the virus IRES-mediated translation (Supplementary Fig. 11c), demonstrating that (GGGGCC)$_{70}$ RAN translation can initiate independent of 5′ cap structure.

We also measured the translation in response to the oxidative stress induced by sodium arsenite via live cell imaging. The stress enhanced translation as early as 7 min, increasing both the number of translating intron RNAs and the intensity of translation sites (Fig. 5e, Supplementary Movie 3). After 30 min of treatment, the number of nascent peptides per translation site became twice as many on average (Fig. 5f). Altogether, the evidence supports that the circular spliced intron, but not the unspliced pre-mRNA, can serve as the template for RAN translation, which is upregulated by stress stimuli.

**The NXF1-NXT1 pathway regulates the circular intronic repeat RNA export**. We next explored how the circular intronic repeat RNA is exported from the nucleus to the cytoplasm. Previously, we identified that the NXF1-NXT1 pathway, modulates the DPR levels produced from a (GGGGCC)$_n$ RAN translation reporter in which the repeat is located in the mRNA[35]. We now tested whether this pathway can also mediate the export of (GGGGCC)$_n$-containing spliced introns. Knockdown of NXF1 by siRNA (Supplementary Fig. 11d) dramatically reduced the number of cytosolic introns compared with the control, whereas the exon number in cytosol was only modestly affected (Fig. 6a, b). The decrease in cytoplasmic intron was not due to reduction in its expression level (Supplementary Fig. 11e), confirming the nuclear export defects. This demonstrates that the repeat-mediated intron export is an active process regulated by NXF1-NXT1 pathway.

If NXF1-NXT1 pathway preferentially modulates the export of the repeat-containing intron, the template of RAN translation, we

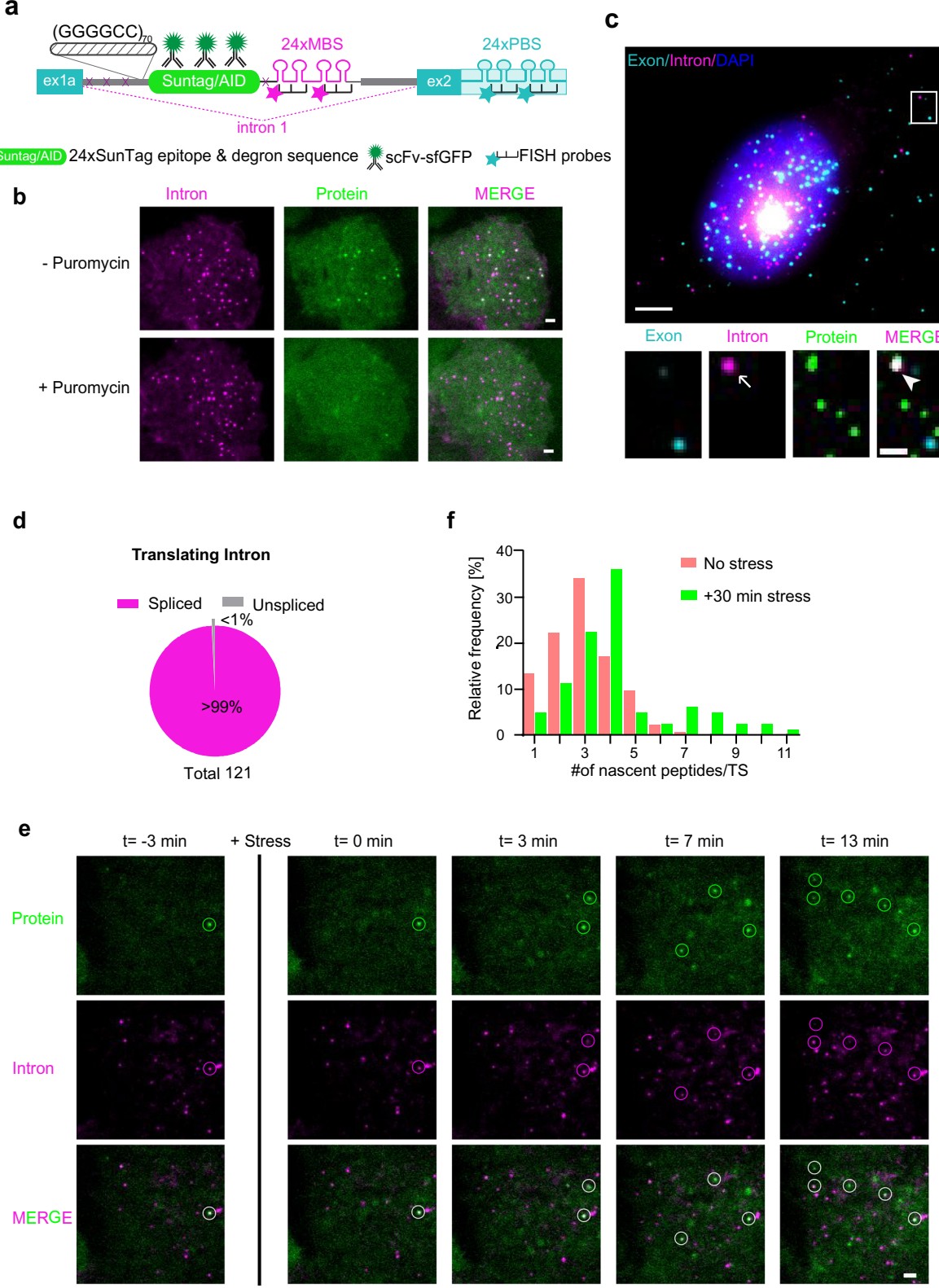

expect that knocking it down will decrease the DPR production and accumulation. We used a previously established dual-luciferase reporter system in the inducible HeLa Flp-In cell line[22]. The Nanoluc Luciferase is generated through RAN translation of *C9ORF72* (GGGGCC)$_n$ repeats (C9R-NLuc in GA- or GP-frame) located in the first intron of *C9ORF72*, and the

Firefly Luciferase (FLuc) in exon 2 is produced by canonical translation (AUG-FLuc) (Fig. 6c)[22]. Using siRNAs to knock down individual components in the NXF1-NXT1 pathway, we observed that C9R-NLuc is reduced more than AUG-FLuc in multiple reading frames (Fig. 6c), suggesting intronic repeat RNA being preferentially exported through this pathway. We further

**Fig. 5 Translation of circular repeat-containing intron. a** Diagram of single-molecule translation imaging reporter for *C9ORF72* repeat-containing intron. The SINAPS reporter lacking the ATG start codon was inserted after the (GGGGCC)$_{70}$ repeats in the GR frame in the splicing construct. The SunTag fluorescence (green, scFv-sfGFP) on the RNA represents the RAN translation signal. The auxin-inducible degron (AID) at the end of the reporter was used to degrade the mature proteins and decrease the background signals[21]. **b** Live cell imaging illustrates that the fluorescence puncta are translation sites. U-2 OS cells stably expressing membrane tethered stdMCP protein[31, 57] were transiently transfected with intronic translation reporter in **a**. Cells were treated with translation inhibitor puromycin (100 μg/mL) at time 0 and imaged at 10 min. Translation was shut down efficiently. Green (protein): SunTag-scFv-sfGFP; magenta (intron): stdMCP-Halotag-CAAX-JF646. Scale bar: 2 μm. Total three independent biological replicates were examined with similar results. **c** Representative smFISH & IF images to measure RNA and translation. Green: protein; magenta: intron; cyan: exon. Arrow: intron; arrowhead: translation site. Scale bar: 5 μm, and 1 μm for zoom in. The protein signals colocalized with the spliced intron (magenta) indicated the translation of the spliced intron. Quantification shown in panel **f** (also see Supplementary Movie 2). **d** Percentage of translating introns colocalized (unspliced) or uncolocalized (spliced) with exons. From a total of 121 translation introns, only 1 showed colocalization with exon signals. **e** Time-lapse images illustrating the upregulation of RAN translation by stress stimuli. U-2 OS cells stably expressing membrane tethered stdMCP protein were transiently transfected with translation reporter for *C9ORF72* repeat-containing intron. Cells were treated with 2 mM sodium arsenite at time 0 and imaged every 30 s for 30 min. Green (protein): SunTag-scFv-sfGFP; magenta (intron): stdMCP-Halotag-CAAX-JF646. Scale bar: 2 μm. Total three independent biological replicates were examined with similar results (also see Supplementary Movie 3). **f** Histogram of translation site intensities measured by the number of nascent peptides on each translation site, which increased upon stress stimuli. Data are quantified from three biological replicates: control (530, 480, 504 cells) and under stress (550, 512, 430 cells). Source data are provided as a Source Data File.

confirmed the effect on endogenous DPR production in C9ORF72-ALS patient-derived induced pluripotent stem cell (iPSC) differentiated neurons (iPSN). NXT1 or NXF1 siRNA transfection in neurons reduced the poly-GP levels measured by the enzyme-linked immunosorbent assay (ELISA) (Fig. 6d, Supplementary Fig. 11f). This supports that NXF1-NXT1 can modulate the endogenous DPR levels in patient cells by regulating the repeat-mediated circular intron export.

## Discussion

The finding of this work is summarized in a working model (Fig. 6e). After transcription, the repeat-containing spliced intron and unspliced pre-mRNA form nuclear RNA granules. Some spliced introns remain in a circular form, are exported to cytoplasm and undergo RAN translation to produce toxic DPR proteins, which is further upregulated by stress stimuli. G-rich repeats contribute to the stabilization of circular lariat, and the NXF1-NXT1 pathway plays an important role in its export.

Microsatellite repeat expansions are linked to ~50 neurological and neuromuscular diseases. While the expansions occur in both coding and noncoding regions of host genes, the repeats located in the intron have more diverse sequences. It was reported that GC-rich intronic expansions can inhibit splicing and trigger host intron retention[30]. The intron-retaining *C9ORF72* transcripts were found to increase in patient postmortem brain and proposed to be the source of both nuclear granules and RAN translation substrates in cytoplasm[26]. Traditional biochemical methods to separate nuclear and cytoplasmic fractions cannot achieve absolute purity, and PCR amplification of intronic fragment does not distinguish whether the intron is spliced out or in the unspliced pre-mRNA. We now used single-molecule imaging approach to directly visualize and precisely quantify different RNA species and their spatiotemporal localization. Although GGGGCC$^{exp}$ modestly inhibits splicing, consistent with previous findings, the more drastic effect is the stabilization of spliced introns. While both unspliced pre-mRNA and spliced intron are found in nuclear granules, only the spliced intron is exported to cytoplasm and subjected to translation.

We found that cytoplasmic *C9ORF72* introns mainly existed in circular form. Recently, thousands of circular RNAs have been identified through high-throughput RNA sequencing. Two types of circular RNAs[36] were recognized: exon-derived circRNAs formed by back-splicing; and intron-derived ones formed by either from escaping debranching (the predominant class)[28,29], or recircularization of full-length intron through an unknown post debranching event[29]. Compared with linear RNAs, circular RNAs are generally more stable, likely due to their resistance to RNA exonucleases[37]. They are also found to be highly abundant and conserved in nervous systems, and dynamically modulated during differentiation and aging[38–40]. The *C9ORF72* GGGGCC$^{exp}$ is located near the 5′ splice site of the first intron. The repeat sequence is close to the branch site during splicing reaction, and likely inhibits the debranching process and stabilizes the intron in the circular form. Analysis of other repeat expansion sequences showed that only the G-rich repeats CGG$^{exp}$, but not the C-rich or AT-rich repeats, enhances the cytosolic accumulation of spliced introns. It is noted that the tetrapeptides produced from CCUG$^{exp}$ located in intron has been reported in DM2 patient postmortem tissues[41]. It is possible that other intronic elements can promote nuclear export, but were not included in our reporter, or the low level of cytosolic intron RNA is sufficient to produce the observed polypeptides. It is therefore important to consider the genetic context for the driving toxicities of different repeat expansion diseases. We also reanalyzed the circular intron sequences reported previously[29]. The transcriptome-wide assessment further revealed that the cytoplasmic circular introns tend to have higher G content and are more structured than the average introns (Supplementary Fig. 12).

The circular RNAs are often considered to be noncoding RNAs, presumably with regulatory roles. Only a few have been functionally elucidated, such as miRNA sponges[42], regulating transcription[43], regulating RNA binding proteins[44], and regulating innate immune responses signaling pathways[45]. There have been reports of translation of circular RNAs, initiating at canonical AUG start codons[46]. However, the level and prevalence of translation at the molecular level is not well understood. As the circular RNA is a covalently closed loop, translation must initiate in a 5′-cap-independent manner. We have previously shown that GGGGCC$^{exp}$ induced RAN translation through cap-independent mechanism[22], and stress stimuli rapidly enhances the RAN translation level[22,23,47]. The single-molecule investigation in this work further demonstrated that the spliced intron is the template of RAN translation. Importantly, it provided a structural explanation for the observed cap-independent translation mechanism. We also measured the number of nascent peptides per translating intronic RNA and the time-lapse changes induced by stress, providing a quantitative measurement of RAN translation dynamics at the single-molecule level.

Different types of RNA molecules are exported from the nucleus into the cytoplasm through the nuclear pore complex (NPC) via various pathways[48]. NXF1-NXT1, in conjunction with transcription-export complex (TREX) and TREX-2, is the

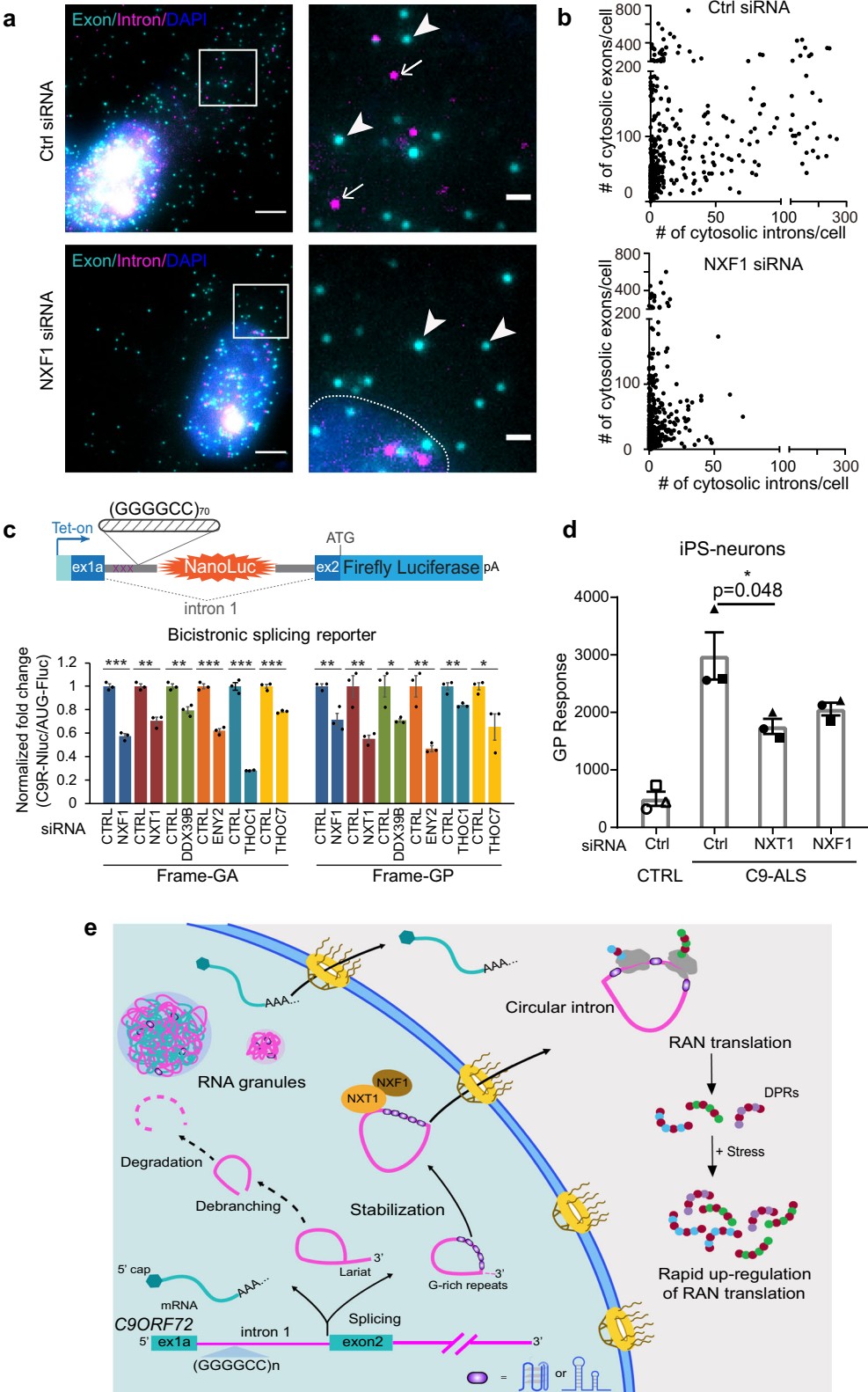

predominant pathway for mRNA export[49]. A subset of mRNAs, rRNAs as well as U snRNAs are exported in a Crm1/Xpo1-dependent manner[50]. Trafficking of circular RNAs, however, is still poorly understood. Recent studies suggest that DDX39B and DDX39A, components of TREX, regulate the exonic circRNA export[51], and cytoplasmic circular RNAs are exported depending on the NXF1/NXT1 system[29]. For repeat-containing circular

introns, we have shown that upon knocking down NXF1-NXT1 pathway, both cytoplasmic intron number and the corresponding RAN translation product are reduced significantly more than the control exonic mRNAs transcribed from the same gene (Fig. 6a–c). This strongly suggests that NXF1-NXT1 pathway plays a pivotal role in regulating the export of this type of circular introns, and has the therapeutic potential to reduce the DPR-

**Fig. 6 The NXF1-NXT1 pathway regulates the circular intronic repeat RNA export. a** Two-color smFISH in $+(GGGGCC)_{70}$ splicing reporter cells transfected with nontargeting control (top) or NXF1-targeting (bottom) siRNA. Magenta: intron (MBS); cyan: exon (PBS); blue: DAPI. Scale bar: 5 μm. The boxes were zoomed and shown on the right, scale bar: 1 μm. Arrow: intron; arrowhead: exon. Dash line: nuclear boundary. Quantification shown in panel **b**. **b** Scatter plot of cytosolic intron vs exon numbers in each cell with either nontargeting control siRNA (top) or NXF1-targeting siRNA (bottom) transfection. Each dot represented one cell (total 418 cells for control and 490 for NXF1 siRNA, from three biological replicates). **c** HeLa Flp-In bicistronic splicing reporter cells (C9R-NLuc is located in the *C9ORF72* intron 1, flanked by C9 exon 1 and 2, pA: poly-A tail)[35] were induced to express translation reporters by doxycycline after two days of siRNA transfection, and luciferase activities were measured after another 24 h. NLuc signals were normalized to FLuc in each sample and the relative expression was compared to nontargeting siRNA control. *$P < 0.05$, **$P < 0.01$, ***$P < 0.001$, two-tailed $t$-test. Data are mean ± SEM. from three biological replicates. The exact $P$ values for frame-GA from left to right are 0.0001, 0.0012, 0.0036, 0.0001, 0.00002, 0.0003. The exact $P$ values for frame-GP from left to right are 0.009, 0.0093, 0.033, 0.0046, 0.0085, 0.0417. **d** C9ORF72-ALS iPSNs were transfected with NXT1, NXF1 or nontargeting siRNA at day 16 post-differentiation. Poly-GP was measured by ELISA after another 16 days of differentiation and maturation. Different shapes of datapoints represent independent cell lines. Data are presented as mean ± SEM. from three control and three C9ORF72-ALS cell lines. *$P < 0.05$ by One-way ANOVA followed by Dunn's post hoc. **e** Working model: G-rich repeats stabilize introns in circular form and mediate the nuclear export. In C9ORF72-ALS/FTD, the circular intron with GGGGCC$^{exp}$ is the template for RAN translation in the cytoplasm, which can be upregulated by stress stimuli. The NXF1-NXT1 pathway plays an important role in the nuclear export of the circular intron, mediated by the GGGGCC$^{exp}$. Source data are provided as a Source Data File.

mediated toxicity. It is possible that specific proteins bind the G-rich repeat sequences/structures to mediate the process, and other pathways may contribute to the export, which requires further exploration.

This study demonstrated that the sequence and/or structure of repeat expansion and the location of the repeats could govern the post-transcriptional processing and the subsequent toxic pathways. The molecular identity of the repeat-containing RNAs need to be carefully examined in order to understand how the expansion contributes to the disease pathogenesis. The single-molecule approach developed here provides an important tool to distinguish the circular RNA from the linear version in situ to delineate their spatiotemporal dynamics. Understanding molecular features of pathological RNA species at the single-molecule level may reveal novel insights into the disease mechanisms and may open new opportunities for therapeutic target design.

## Methods

**Plasmids**. For the C9ORF72 splicing reporter, the sequences including *C9ORF72* exon 1a, 50nt of exon2, and around 200nt intron 1 from each exon–intron junction were synthesized (Genewiz) and cloned into pCAG vector via HindIII (DNA blunting at the HindIII end) and XhoI. We included HindIII and BamHI sites inside the synthesized sequence close to the repeat expansion location for downstream cloning of repeat sequences. Next, the first 300nt of the Fluc coding sequence was inserted into pCR4-24×PP7SL[14] (Addgene Plasmid #31864) via SpeI and BamHI. The fused FLucN-24×PBS was then cut out by XhoI and BglII (Blunt) and inserted downstream of *C9ORF72* exon2 via XhoI and NsiI (Blunt) to generate pCAG-C9splicing-exFLucN-24×PBS. To make the pCAG-(GGGGCC)$_{70}$-24×MBS mRNA expression plasmid, 24×MBSV5 was PCR amplified from puc57-24×MBSV5 template[15] and inserted downstream of (GGGGCC)$_{70}$ via NsiI and SacI in the pCAG-(GGGGCC)$_{70}$ vector[35]. The FLAG-24×SuntagV4 was cut out from pUbC-FLAG-24×SuntagV4-oxEBFP-AID-baUTR1-24×MBSV5-Wpre[21] (Addgene Plasmid #84561) by NotI and FspI and inserted into pCAG-(GGGGCC)$_{70}$-24×MBS via NotI and NsiI (Blunt) between (GGGGCC)$_{70}$ and 24×MBS. The ATG start codon was removed and a frame shift was introduced by replacing the region between NotI and AflII with a linker sequence AGATTA-CAAGGACGACGACGATAAGGGCGGACCGGGTGGATCTGGAGGTGGAG GTTCTGGAGGAGAAGAACTTTTGGACAAGAATTATCATCTTGAGAA CGAAGTGGCTCGT for GA frame and GATTACAAGGACGACGACGATAA GGGGCGGACCGGGTGGATCTGGAGGTGGAGGTTCTGGAGGAGAAGAA CTTTTGAGCAAGAATTATCATCTTGAGAACGAAGTGGCTCGT for GR frame). The (GGGGCC)$_{70}$-24×Suntag-24×MBS fragment was subsequently cut out by HindIII and BglII and inserted into the intron of pCAG-C9splicing-exFLucN-24×PBS vector at the HindIII and BamHI sites to generate pCAG-C9splicing-in(GGGGCC)$_{70}$-24×Suntag-24×MBS-exFLucN-24×pbs (Suntag in GA or GR frame). We removed the 24×Suntag by cutting the above construct with NotI and NheI, blunting the ends and self-ligating to generate the C9ORF72 splicing reporter pCAG-C9splicing-in(GGGGCC)$_{70}$-24×MBS-exFLucN-24×PBS; or removed (GGGGCC)$_{70}$-24×Suntag by cutting with HindIII and NheI, blunting the ends and self-ligating to generate the no-repeat control C9ORF72 splicing reporter pCAG-C9splicing-in24×MBS-exFLucN-24×PBS. To delete the 5′ss, the entire fragment containing Intron-(GGGGCC)$_{70}$-24XMBS-Intron-exFLucN-24XPBS was cut out

from the C9ORF72 splicing reporter by HindIII and SacI, and cloned into pCAG vector.

For splicing reporters with other repeats in the *C9ORF72* gene context, the (GGGGCC)$_{70}$ repeats was replaced by a fragment containing multiple cloning sites (GCTAGCCAATTGAAGCTTATGCATGCGGCCGCGCTGAGGTGTACA) through Gibson assembly (New England Biolabs, E2611S). (CGG)$_{98}$ was cut from pGLACTE-PseudoHT51 (Addgene Plasmid #99257) by HindIII and NheI (blunt) and inserted into the multiple cloning sites of the splicing reporter by HindIII and NotI (blunt). (CCCCGG)$_{70}$ was inserted into the HindIII and NsiI sites. (CCTG)$_{240}$ was cut from DT240 (Addgene Plasmid #80414) by SalI (Blunt) and HindIII and inserted into the NheI(blunt) and HindIII sites. (ATTCT)$_{73}$ was synthesized (Genewiz) and cloned into the splicing reporter via HindIII and NotI.

For the ZNF9-(CCTG)$_{240}$-splicing reporter in the genetic context of *ZNF9* itself, the sequences including 100nt ZNF9 exon 1, 49nt of exon2 and around 200nt intron 1 from both 5′ and 3′ exon–intron junction were synthesized (Genewiz) and cloned into pCAG vector via HindIII and NsiI. The FLucN-24×PBS was cut from pCAG-C9splicing-in24×MBS-exFLucN-24×PBS and inserted after the exon2 by XhoI + SacI. The (CCTG)$_{240}$ was cut from DT240 by SalI (Blunt) and BamHI and inserted into the the above construct by EcoRV and BamHI. At last, PCR amplified 24×MBS was inserted via AgeI and AvrII.

For the ATXN10-(ATTCT)$_{73}$-splicing reporter in the genetic context of *ATXN10* itself, the sequences including 69nt ATXN10 exon9, 45nt of exon10 and around 200nt intron9 from both 5′ and 3′ exon–intron junction were synthesized (Genewiz) and cloned into pCAG vector via BbvCI and NsiI. The FLucN-24×PBS was cut from pCAG-C9splicing-in24×MBS-exFLucN-24×PBS and inserted downstream of exon2 by XhoI + SacI. (ATTCT)$_{73}$-24×MBS was cut from the previous C9-(ATTCT)$_{73}$-splicing reporter and inserted in the ATXN10 reporter by NheI and AvrII.

For the mCherry-24xMBS plasmid, we used PCR fragment of mCherry with appended AgeI site to replace the DsRed-IRES-GFP in the phage-ubc-RIG lentiviral vector[52], cut at NotI and ClaI. 24×MBSV5 was inserted with AgeI and ClaI sites after the stop codon and before the WPRE element.

To generate the bicistronic reporters for in vitro transcription, a 200 bp ORF was PCR amplified from the 3′ end of the FLuc sequence containing multiple stop codons in all three reading frames. The ORF was inserted before NLuc or C9R-NLuc in the pcDNA3.1-NLuc or pcDNA3.1-(GGGGCC)$_{70}$-NLuc constructs (both frame-GA and frame-GP)[22]. A 600 bp random nonstructured sequence was inserted between the 200 bp ORF and NLuc by EcoRV and NotI to generate the pcDNA3.1-ORF1-Neg-NLuc. For IRES-positive controls, the EMCV or CrPV IRES was PCR amplified from templates[53] and cloned between ORF1 and NLuc to generate pcDNA3.1-ORF1-EMCV/CrPV IRES-NLuc.

**Cell culture and transfection**. U-2 OS (American Type Culture Collection HTB-96) and HEK293T (American Type Culture Collection CRL-1573) cells were grown in DMEM supplemented with 10% (v/v) FBS, 100 U/ml penicillin, and 100 μg/ml streptomycin (Sigma–Aldrich). To generate polyclonal stable cell lines, U-2 OS cells (U2PA stably expressing stdMCP-Halotag and stdPCP-stdGFP[15] for splicing reporter, U2PA stably expressing stdMCP-Halotag and scFv-sfGFP[21] for Suntag translation reporter) were transfected with linearized reporter constructs by nucleofection (Lonza nucleofector 2b, VACA-1003), followed by 200 μg/ml hygromycin selection for 20 days. HEK293T cells were transfected by TransIT®-LT1 Transfection Reagent (Mirus Bio). ALS patient-derived fibroblast cell lines (C9-2, C9-5, C9-6) and healthy control fibroblast cell lines (Ctrl-1, Ctrl-2) are gifts from Dr. John Ravits[25] were grown in DMEM supplemented with 20% (v/v) FBS, 100 U/ml penicillin, 100 μg/ml streptomycin, 2 mM Glutamine and 1% (v/v) nonessential amino acids. Patient lymphoblast cells were obtained from Coriell Institute and grown in RPMI1640 supplemented with 10% (v/v) FBS, 2 mM L-

Glutamine, 100 U/ml penicillin, and 100 μg/ml streptomycin. The cell line information is listed in Supplementary Table 1.

Lipofectamine® RNAiMAX (Invitrogen) was used to transfect siRNAs and ASOs. ON-TARGETplus pooled siRNAs of *NXF1* (GE Dharmacon, L-013680-01-0005), *NXT1* (L-017194-00-0005), *DDX39B* (L-003805-00-0005), *ENY2* (L-018808-01-0005), *THOC1* (M-019911-01-0005), *THOC7* (M-014575-01-0005), and nontargeting control (D-001810-10-05) were transfected at 25 nM and cells were harvested after 3 days. Nontargeting ASO (CCTTCCCTGAAGGTTCCTCC) and the *C9ORF72*-targeting ASO (GCCTTACTCTAGGACCAAGA)[25] were transfected at 25 nM, and cells were collected after 2 days.

**Western blot**. For immunoblotting, goat anti-mouse or anti-rabbit IgG HRP-conjugated (1:5000, GE healthcare, #NA934V and #NA934V) was used along with chemiluminescent detection reagents (Thermo Scientific). The primary antibodies included NXF1 (1:1000, Bethyl Laboratories, #A300-914A), β-actin (1:1000, Cell Signaling Technology, #3700).

**iPSN differentiation, transfection, and poly-GP ELISA**. iPSC lines were obtained from Answer ALS and listed in Supplementary Table 1. The iPSCs were maintained and differentiated into spinal motor neurons according to the publicly available 18-day 'diMNs' (direct iMN) protocol (http://neurolincs.org/pdf/diMN-protocol.pdf)[54,55]. At day 12 post-differentiation, iPSNs were dissociated and seeded in 24-well plate. 100 μL S3 medium containing 1 μL 100 μM siRNA and 3 μL Lipofectamine® RNAiMAX (Invitrogen) was incubated at room temperature for 10 min before adding onto iPSNs on day 16. S3 medium was half-changes every 3 days till day 32. The cell pellet was lysed in ELISA lysis buffer (50 mM Tris pH7.4, 200 mM NaCl, 1% Triton X-100, 5 mM EDTA, 0.5% SDS, 1 mM DTT, and proteinase inhibitor) and the protein concentration was quantified by BCA assay (Pierce). Duplicates of each sample were used for GP ELISA by a technician blinded to the experimental groups as described[35].

**RNA extraction and RT-PCR**. RNA was extracted from cells or tissues by Trizol (Invitrogen, 15596018) and treated with RQ1 DNase I (Promega). The patient postmortem sample information is listed in Supplementary Table 2. cDNA was synthesized by High-capacity cDNA reverse transcription kit (Applied Biosystems, 4368813) using random hexamers. Regular PCR reaction was performed using LiTaq DNA polymerase (Lifesct, M0024). Normal qPCR reactions were performed with three biological replicates for each group and two technical replicates using the Power Up SYBR Green Master Mix (Applied Biosystems, A25776). The data were analyzed using the CFX96 optical system software (Bio-Rad; ver. 1.1). Expression values were normalized to *GAPDH* mRNA. The endogenous *C9ORF72* intron 1 and mRNA levels were detected by TaqMan gene expression assay using TaqMan fast advanced master mix (Applied Biosystems, 4444557), and normalized to β-actin mRNA. Custom primer sequences are listed in Supplementary Table 3. Human *ACTB* primer set (Hs.PT.39a.22214847) was purchased from Integrated DNA technologies.

For RNase R treatment, 10 μg of DNase I digested total RNA was incubated at 37 °C for 15 min with 15U RNase R (Lucigen, RNR07250) to degrade liner RNA, and subsequently purified by Quick-RNA Microprep Kit (Zymo research, R1050). One microgram of RNA before or after RNase R treatment was used as the template for cDNA synthesis.

**In vitro transcription and translation**. The pcDNA3.1-ORF1-Neg-NLuc, pcDNA3.1-ORF1-IRES-NLuc, and pcDNA3.1-ORF1-(GGGGCC)₇₀-NLuc construct (both frame-GA and frame-GP) were linearized with XhoI digestion, and served as the DNA template to synthesize RNA using MEGAscript T7 transcription kit (Ambion, AM1333). In vitro transcribed RNAs were purified by Trizol (Invitrogen, 15596018). One microgram of RNA was used for in vitro translation using Flexi Rabbit Reticulocyte Lysate System (Promega, L4540). Five microliters of completed translation reaction were used to measure NLuc activity by Nano-Glo dual-luciferase reporter assay system (Promega, E1910).

**Nanopore sequencing**. For nanopore sequencing of poly(A) RNA from (GGGGCC)₇₀-splicing reporter cells, the poly(A) mRNA was purified from total RNA by NEBNext® Poly(A) mRNA Magnetic Isolation Module (E7490L). The sequencing samples were prepared using the Direct RNA Sequencing Kit (Oxford Nanopore Technologies, SQK-RNA002). ~500 ng of poly(A) RNA was reverse transcribed with SuperScript IV (ThermoFisher), ligated to the 1D sequencing adapter, and loaded onto the GridION flow cell (Oxford Nanopore Technologies, FLO-MIN106). The sequencing was run for 72 h using the MinKNOW software (ver. 20.06.9) with default parameters (e.g., initial bias voltage of −180 mV). The reads were base-called in real time on the GridION instrument using Guppy with a minimum q score of 7 (ver. 4.0.11), which yielded a final count of 1.52 million reads. The base-called reads were then aligned to the reporter pre-mRNA sequence (including a flanking 100 bp vector sequence upstream and downstream of the transgene) using minimap2 (ver. 2.17, https://academic.oup.com/bioinformatics/article/34/18/3094/4994778) in the nanopore direct RNA sequencing splice alignment mode (-ax splice) with a 14-nt seeding window (-k14), a forced directionality (-uf), a 10,000-nt minimizer window cutoff (-g10000), and a lowered penalty of 5

for noncanonical splice junctions (-C5). Only primary mapping reads with a MAPQ higher than 12 were used for analyzing the presence of various molecules in the experiments, which identified 171 reads on the reporter transgene in stable U-2-OS cell line and 6838 reads in transfected HEK293T cells. Due to the repetitive sequences in 24×PBS, some reads were mapped incompletely and showed non-existing gaps in the alignment. We therefore used the EMBOSS Needle tool (https://www.ebi.ac.uk/Tools/psa/emboss_needle/) as an alternative method, which partially improved the alignments.

**MiSeq library preparation and data analysis**. We generated MiSeq libraries to sequence the *C9ORF72* circular intron and splice junctions of exon 1-2. For circular intron, first-strand cDNA was produced by TGIRT-III reverse transcriptase (InGex, TGIRT10) using 1ug of RNase R-treated RNA. Library was prepared by two rounds of PCR amplification of the fragment across the potential 5′ splice site-branch site junction. For mRNA, first-strand cDNA was synthesized by High-capacity cDNA reverse transcription kit (Applied Biosystems, 4368813). Library was also prepared by two rounds of PCR amplification of mRNA exon 1-2. All primer sequences are listed in Supplementary Table 4. PCR was performed using Phusion High-Fidelity DNA Polymerase (Thermo Fisher Scientific, F530S). The PCR product was resolved on 2% agarose gel and purified by GeneJET Gel Extraction Kit (Thermo Fisher Scientific, K0692) and then subjected to pooled single-end 300nt MiSeq. Sequencing reads were mapped to assembled *C9ORF72* reference sequences (with candidate mRNA isoforms and circular RNAs) using BWA-MEM for mRNA and Minimap 2 (ver. 2.17) for circular intron. The reads that do not cross the splice site-branch site junction were removed by SAMTools.

**RNA smFISH and immunofluorescence**. The RNA single-molecule FISH (smFISH) using 20mer DNA oligo probes was adapted from refs. [17,56] and described in detail[56]. In brief, DNA oligoes were ordered from Integrated DNA Technology and labeled in house with Cy3, Cy5, or Cy7 NHS ester (Lumiprobe 11020, 13020, 15020)[56]. Cells were seeded on 18 mm coverslips coated with 30 μg/mL rat tail collagen I (Gibco, 2052954), and cultured overnight. After fixation with 4% paraformaldehyde and permeabilization with 0.1% of Triton, cells were incubated with 50–100 nM probes in hybridization buffer for 3 h at 37 °C. The unbound probes were washed away with 10% formamide and the coverslips were mounted on microscope slide using ProLong Diamond Antifade Mountant containing DAPI (Thermo Fisher Scientific, P36962) for nucleus staining. For patient fibroblast, the cells were treated with 70% of ethanol overnight after fixation and before permealization. The RNA FISH probe sequences are listed in Supplementary Data 1.

Cells used for single-molecule translation assay were treated with 500 μM IAA overnight. The protocol for smFISH immunofluorescence (IF) experiments was detailed in ref. [57]. Briefly, cells were fixed with 4% paraformaldehyde for 10 min and blocked with 50 mM Glycine in 1× phosphate buffer saline (PBS) at room temperature. After permeablization with 0.1% Triton at room temperature, cells were incubated with 100 nM probes targeting RNA of interest and primary antibody targeting sfGFP (Aves Labs, GFP-1010) at 37 °C for 3 h. After washing away unbound primary antibody and probes, cells were washed twice with secondary antibody (Goat anti-Chicken IgY (H+L) Alexa Fluor 647, Thermo Fisher Scientific, A21449) at 37 °C for 20 min. Finally, the coverslip was washed with PBS before mounting on microscope slide as mentioned in smFISH protocol.

For smFISH with RNase R treatment, cells were fixed with Methanol-Acetic Acid (Fisher A38-212) mixture (4:1) for 20 min at −20 °C after a quick wash with ice cold PBS. It is critical not to use paraformaldehyde for fixation because it may crosslink protein and RNAs. Next, cells were washed with 1×PBS, incubated with 1× RNase R buffer for 5 min at room temperature, and treated with RNase R (1 U/50uL in 1× RNAse R buffer) at 37 °C for 15 min. Cells were washed with 1× PBS supplemented with 5 mM MgCl₂ for 5 min and subsequently proceeded the same as normal smFISH protocol, with 50 nM probes targeting *POLR2A* (Cy3), MBSV5 (Cy5), and PBS (Cy7).

**Fluorescence microscopy**. The fixed samples were imaged on an automated inverted Nikon Ti-2 wide-field microscope equipped with ×60 1.4NA oil immersion objective lens (Nikon), Spectra X LED light engine (Lumencor), and Orca 4.0 v2 scMOS camera (Hamamatsu). The live cell experiments were performed on a custom microscope built around Nikon Ti-E stand. The excitation was through HTIRF (Nikon) with an LU-n4 four laser unit (Nikon) with solid state lasers with wavelengths 405, 488, 561, and 640 nm. The main dichroic was a quad band dichroic mirror (Chroma, ET-405/488/561/640 nm laser quad band set for TIRF applications). The imaging was done through the ×100 1.49NA oil immersion objective (Nikon). To achieve simultaneous 3-color imaging, we used a TriCam light splitter into three separate EMCCD cameras (Andor iXon Ultra 897) with ultraflat 2 mm thick imaging splitting dichroic mirrors (T565LPXR-UF2, T640LPXR-UF2). A band pass emission filter was placed in front of each camera, respectively (ET525/50 m, ET595/50 m, and ET655lp). The microscope was also equipped with an automated XY-stage with extra fine lead-screw pitch of 0.635 mm and 10 nm linear encoder resolution and a Piezo-Z stage (Applied Scientific Instrumentation) for fast Z-acquisition. The whole microscope was under the control of Nikon Elements for automation.

**Live cell imaging**. Cells were seeded in a 35 mm dish with 20 mm micro well #1.5 cover glass bottom (Cellvis, D35-20-1.5-N) and grown overnight. Two hours prior to imaging, cells were incubated for 1 h with 100 nM Halo dye (JF646)[20] and washed for 1 h in the incubator. The media was changed to Leibovitz's L-15 media supplemented with 10% FBS and the sample was kept at 37 °C with humidity control with a Tokai Hit stage top incubator during imaging.

For translation imaging, the pCAG-C9splicing-in(GGGGCC)70-24×Suntag-24×MBS-exFLucN-24×PBS reporter was transiently transfected into U2PA-stdMCP-halo-CAAX cell line 24 h before imaging. For stress response, the sample was initially imaged every 30 s for 3 min, then treated with 2 uM arsenite and imaged every 30 s for 40 min to capture changes in translation. For translation inhibition, the sample was initially imaged every 30 s for 3 min, then treated with 100 μg/mL puromycin for 10 min and imaged afterwards.

**Image quantification and analysis**. To determine the coordinates and fluorescence amplitudes of smFISH and translation site signals, images were analyzed using the Matlab software FISH Quant[58]. In brief, single-molecule spots for RNA in cytoplasm and nucleus were fitted to a 3D Gaussian function after background subtraction to extract the centers and amplitudes of the fluorescent spots. To correct chromatic aberration, we imaged 0.1 μm TetraSpec beads (Thermo Fisher, T7279) and calibrated an affine transformation matrix to register beads positions in different channels[59]. The position of RNA and protein was adjusted with the transformation matrix for the colocalization analysis between intron and exon or between RNA and translation site. A threshold of 200 nm was applied as a criterion for colocalization. Next, we determined the brightest IF spot within the distance threshold of an intron. If its intensity was brighter than a single mature SunTag protein, we deemed it as a translation site. Finally, the distance between the identified translation sites and the closest exons were measured and compared with the threshold (300 nm) for colocalization criterion.

The positions and intensities of RNA foci were determined with the "Transcription Site" modality in the FISH Quant software. We applied a threshold of six times of single-RNA intensity to be considered as foci. The number of single RNAs in each focus was determined by dividing its integrated intensity by the corresponding single RNA one.

The live imaging analysis was performed using existing packages or custom build programs in Matlab (MathWorks). Single particles of RNA or translation sites were identified with Airlocalize[60] and the particle positions between frames were tracked by u-track[61]. Tracks longer than five frames were further analyzed. The diffusion coefficient for each track was calculated using mean-square-displacement (MSD) with the first seven displacements. RNAs with diffusion coefficient larger than 0.01 μm$^2$ /s were classified as freely diffusing.

**RNA G-content calculation**. The circular intron sequences in HeLa cells were obtained from ref.[29]. The genome-wide intron sequences were extracted with the R BioConductor GenomicFeatures package (https://bioconductor.org/packages/release/bioc/html/GenomicFeatures.html) using the RefSeq database (hg19 version from January 10, 2020; downloaded from UCSC). Only constitutive introns that contain no alternative exons were selected for analysis. Intron G content was calculated using a custom built script in Matlab (MathWorks). In brief, for each circular intron, the number of G nucleotide was divided by the full length of the sequence. For genome-wide introns, we removed the last 30 nucleotides from the 3′ splice site (the average distance between branch site and 3′ splice site) and selected the rest sequences longer than 25nt. To calculate the percentage of G at the single nucleotide level, we aligned all introns starting from the 5′ or from 3′ splice sites, the total number of G nucleotide in each position is divided by the total number of sequences considered. For genome-wide introns, we selected ones that full length was longer than 25nt for this calculation.

**RNA secondary structure prediction**. RNA secondary structure was predicted by the "RNAfold" function from ViennaRNA package. The "gquad" modifier was used to incorporate G-Quadruplex formation into the structure prediction algorithm to calculate the minimum free energy (ΔG) of the RNA secondary structure. The ΔG value was divided by the length of the RNA sequence to calculate the minimum thermodynamic free energy (−ΔG /nt). For genome-wide intron secondary structure prediction, introns between 100nt and 600nt were selected and the last 30nt from the 3′ splice site (the average distance between branch site and 3′ splice site) were trimmed.

**Reporting summary**. Further information on research design is available in the Nature Research Reporting Summary linked to this article.

## Data availability
The data supporting the findings of this study are available from the corresponding authors upon reasonable request. The MiSeq and Nanopore sequencing data generated in this study have been deposited to NCBI SAR under accession code (PRJNA660882). Source data are provided with this paper.

## Code availability
The analysis code that supports the findings of this study is available in GitHub[62] (https://github.com/binwulab/CircularIntron).

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

## Acknowledgements

We thank Target ALS Human Postmortem Tissue Core to provide us postmortem samples. We thank the Sun and Wu lab members for helpful discussion. This work is supported by Johns Hopkins University School of Medicine Discovery Fund Synergy Award, National Institute of Health (RF1NS113820 to S.S. and B.W.; R01NS107347 to S.S.; K08NS104273 to L.R.H.; NS099114, NS094239, AG057623 to J.D.R.), National Science Foundation (MCB 1817447 to B.W.), Department of Defense (J.D.R.), Pew Charitable Trust (00030601 to B.W.), Target ALS (S.S.), ALS Association (S.S.) and the Packard Center for ALS Research. Z.Z. was a recipient of the Milton Safenowitz Post-Doctoral Fellowship from the ALS Association. M.J.L. and D.G.B. were supported by NIH Training Grant (T32 GM008403), N.M.L. was supported by NIH Training Grant (T32 GM007445).

## Author contributions

S.W., M.J.L., B.W., and S.S. contributed to the overall design and interpretation of the study, and wrote the manuscript with input from the other authors. S.W., M.J.L., Z.Z., D.D., N.M.L., and W.T. performed most molecular and cellular biology experiments under the mentorship of S.S. and B.W. D.G.B. and B.H. helped with the bioinformatics analysis of the high-throughput sequencing data. L.R.H. and J.D.R. performed GP ELISA assay and analysis. L.W.O. advised on patient samples and provided key reagents.

## Competing interests

The authors declare no competing interests.
