## [Peer Review File · Nature Communications]

REVIEWER COMMENTS

Reviewer #1 (Remarks to the Author):

In C9orf72 ALS/FTD, the hexanucleotide GGGGCC repeats are translated into GA, GP, and GR dipeptide repeat proteins. However, the exact form(s) of RNA that undergoes translation is still unclear. In the current study, Wang and colleagues generated multiple RNA reporter cell lines, and used both FISH and live-cell imaging in an attempt to elucidate the structure/composition of the translated RNA. Based on observations in both reporter cell lines and patient fibroblasts, they argue that the excised intron lariats are stabilized, exported, and even translated. This is an extraordinary claim. However, several key analyses were based on unestablished assumptions, and are therefore open to alternative interpretations.

Major concerns

1. Throughout the manuscript, analyses using the reporter constructs were based on the assumption that only three products would be generated: an unspliced RNA, a spliced mRNA (exon), and an intron lariat. A single RT-PCR experiment with three pairs of primers was performed to validate this construct. However, reporter constructs are notoriously prone to misprocessing due to cryptic splice sites. Any cryptic splicing that excluded the 24xPBS region would lead to false positive signals misinterpreted as the excised intron. To rule out these possibilities, the authors should perform RNA-seq to confirm that only the expected transcript isoforms are generated.
2. Another important validation for the 24xMBS puncta as excised introns would be the requirement for each of the two splice sites. This is currently missing.
3. In the patient fibroblast analysis (Fig. 2), no validation was provided to ensure the specificity of either intron or exon probes.
4. Two peculiarities were unexplained in Fig. 2. First, the intron probes did not detect any RNA foci. Second, nuclear intron signals were similar between control and C9 patient cells, which contradicted Fig. 1e.
5. Attempting to show that the intron puncta are in a lariat form, the authors treated fixed cells with RNase R, and found that the cytoplasmic intron puncta were somewhat resistant. Even for extracted RNAs, RNase R resistance could be caused by reasons other than circularity (e.g., PMID: 31269210). Here, the interpretation is further confounded by the presence of all of the proteins bound RNAs, which might also interfere with RNase R digestion. As a result, even the spliced exonic RNA was not completely removed (Fig. 3c). A definitive method to test circularity is by 2D denaturing gels, which have been used to isolate intron lariats (PMID: 27473169).
6. It was not explained why the stabilized, nuclear introns were efficiently digested by RNase R (Fig. 3d).
7. The translation analysis in Fig. 5 was not sufficiently explained. What percent of cytoplasmic intron RNAs are translated? Fig. 5e shows that multiple nascent chains were found on each RNA, which is surprising considering that RAN translation is known to be inefficient.
8. Perhaps the most provocative claim is that the capless intron lariats can be translated, which would require an IRES-like initiation mechanism. The authors should test the C9 intron sequence for IRES activity with in vitro transcribed RNAs to avoid common caveats of cellular IRES assays (PMID: 20576611).
9. A previous study from the same group (PMID: 31587919) has shown that NXF1 and NXT1 also

promote the nuclear export of linear GGGGCC repeat RNAs. Therefore, it appears to be the repeat sequence, rather than circularity that determines its export pathway.

Minor issues

1. Results in Fig. 3e is expected by design. They do not show that intron puncta are predominantly lariats.
2. The authors found that intron stabilization and export was specific to G-rich repeats including GGGGCC and CGG, but not to other disease-associated repeats. However, CCTG repeats in DM2 are also intronic and RAN-translated. The authors should discuss the apparently distinct RAN translation mechanisms between these two repeat expansion mutations.
3. A typo ("Raipd") in Fig. 6e.
4. A space is missing in Page 3 Line 22.

Reviewer #2 (Remarks to the Author):

In this excellent study by Wang et al, the authors use advanced imaging methods to investigate the localization and translation of the circular repeat-containing intronic RNA C9ORF72. Using a clever reporter construct incorporating the intron and neighboring exons, they exploit the MS2/PP7-RNA reporter system as well as the SunTag-reporter system to investigate the spatiotemporal regulation of repeat RNA localization and translation. They find the hexanucleotide repeat expansion to be within a circularized intron, that is exported from the nucleus and translated. Validation of these findings in C9ORF72-ALS patient samples and analysis of the export pathway is further strengthening the paper. The paper is very well written and the data are well presented making it easy to follow the story. Overall, the results are novel, original, very convincing and the paper is a significant and major advance for the field. Also, I would like to stress that the results have a significant and high impact for a large and interdisciplinary community justifying publication with high priority.

There is a couple of points the authors could consider to improve their story: Whereas the paper is well written, the figure legends would certainly benefit from another editing attempt. Clearly, better explanation in figure legends / annotations within figures would further improve readability and do more justice to the impressive data.

In addition, the SunTag portion could be improved (Fig. 5). While the data from the SunTag experiment look very convincing, a control including a translation inhibitor is crucial in my opinion to exclude possible artefacts due to aggregation of SunTag-reporters without active translation. Possibly, the authors have done this already, but I suggest incorporating this data into the main figure.

Finally, Fig. 1 could be improved. For improved understanding of Fig. 1b, it would help to include a depiction of the smFISH probes and possibly exclude the MS2 coat protein and PP7 coat protein labelling, since the analysis in Fig. 1 was performed solely using smFISH. Furthermore, the y-axis title in Fig. 1c is misleading, I guess displayed are the number of nuclear RNA granules normalized to the total number of RNA foci per cell.

Reviewer #3 (Remarks to the Author):

C9ORF72 hexanucleotide GGGGCC repeat expansion is the most common cause of ALS and FTD. Repeat-containing RNA mediates toxicity through nuclear granules and dipeptide repeat (DPR) proteins produced by repeat-associated non-AUG translation. In this manuscript, the authors present the data suggesting that the splicing intron with G4C2 repeats is stabilized in a circular form, which

could serve as the translation template. They further showed that the NXF1-NXT1 pathway plays an important role in the nuclear export of the circular intron and modulates toxic DPR production. Although the results are potentially interesting, however, all the analyses were performed in vitro and the main conclusion was based on in vitro overexpression system. Additional in vivo and functional data are needed to support the authors' conclusion. Below are some specific comments:

1. The authors did demonstrate the presence of circular intron in cytoplasm, however, it is unclear whether linear intron containing G4C2 repeat is present in cytoplasm. If it does, what's the ratio between linear and circular forms, particularly the transcripts produced from endogenous locus? The linear intron RNA could potentially be present in granules, which would make it difficult to detect using the techniques employed in this manuscript.
2. Instead of using patient fibroblast cells, a disease-relevant cell type(s) should be examined instead.
3. The authors suggested that they tested expanded FMR1 CGG repeat. The size of CGG repeat that they used is 34, which is the normal allele of FMR1.
4. The results of NXF1-NXT1 pathway analysis are interesting, however, it was only done in vitro. Could NXF1-NXT1 pathway could modulate the G4C2 repeat toxicity in vivo?

Response to reviewers' comments

We would like to thank all the reviewers for the very constructive comments and suggestions, which help us significantly improve this manuscript. We extensively modified the manuscript with more experiments and controls. The major modifications were marked with yellow highlight in *Results*, *Method* and *Figure legend*.

Reviewer #1

In C9orf72 ALS/FTD, the hexanucleotide GGGGCC repeats are translated into GA, GP, and GR dipeptide repeat proteins. However, the exact form(s) of RNA that undergoes translation is still unclear. In the current study, Wang and colleagues generated multiple RNA reporter cell lines, and used both FISH and live-cell imaging in an attempt to elucidate the structure/composition of the translated RNA. Based on observations in both reporter cell lines and patient fibroblasts, they argue that the excised intron lariats are stabilized, exported, and even translated. This is an extraordinary claim. However, several key analyses were based on unestablished assumptions, and are therefore open to alternative interpretations.

Major concerns

1. Throughout the manuscript, analyses using the reporter constructs were based on the assumption that only three products would be generated: an unspliced RNA, a spliced mRNA (exon), and an intron lariat. A single RT-PCR experiment with three pairs of primers was performed to validate this construct. However, reporter constructs are notoriously prone to misprocessing due to cryptic splice sites. Any cryptic splicing that excluded the 24xPBS region would lead to false positive signals misinterpreted as the excised intron. To rule out these possibilities, the authors should perform RNA-seq to confirm that only the expected transcript isoforms are generated.

Response 1: We thank the reviewer for raising this issue, critical for interpreting our experimental results. To address this question, we carried out RNA-seq experiment suggested by the reviewer. Due to the multiple repeat sequences (GGGGCC^{exp}, 24xMBS, 24xPBS) in the reporter, it is difficult to accurately characterize the isoforms using short-read sequencing methods. Therefore, we performed Nanopore long-read sequencing directly on poly(A) enriched mRNAs to ascertain different RNA isoforms from the reporter transgene (**Figure S2**). From the 1.53 million raw sequences, we identified 171 reads that can be aligned to our reporter construct with high confidence (Methods). Among these reads, 24 mRNA sequences span from the 5' (Exon 1a) to the expected poly(A) sites (Group 1). The exon-exon junction site agreed with the isoform we identified from the intron and mRNA amplicon sequencing experiment (Figure S6d). Among these 24 sequences, there were 2 mRNAs missing significant chunk of PP7 sites (*marked)(<4%), but these molecules will be invisible in the imaging experiment. There were 101 molecules that were mapped to the 3' end of the construct with different 5' ends (Group 2). This is most likely due to truncated reading, a common problem in the Nanopore sequencing (read in the 3'→5' direction). Importantly, all these reads contained PP7 regions. Surprisingly, we found a small group of previously unidentified short reads starting from Exon 1a and ending either inside intron 1 (before the repeats) or in exon 2 (before the PP7 binding sites) (Group 3), agreeing with the reviewer's intuition that the reporter transgene might have misprocessing. In this case, weak alternative

poly(A) sites were probably recognized. However, these transcripts contained neither MS2, nor PP7 binding sites, therefore invisible to our imaging experiment and not interfering with our conclusions. We also identified 2 molecules (1%) with a small segment of MS2 (not full-length, hard to be visible at single molecule level) and some random sequences that cannot be mapped correctly to our construct (Group 4), which might be due to errors during the stable integration or sequencing error. **In summary, the Nanopore long-read sequencing of poly(A)-enriched RNAs did not identify transcripts containing 24×MBS, consistent with the fact that the intronic 24×MBS sequence is spliced out and not conjugated with poly(A). Mature poly(A) mRNAs containing 24×PBS did not have 24×MBS, therefore consistent with the imaging results that the exported introns are not in the pre-mRNA. This validates that the predominant MS2 signal in the smFISH experiment represents the spliced intron, not due to misprocessed mRNA containing 24×MBS with spliced-out 24×PBS.**

2. Another important validation for the 24xMBS puncta as excised introns would be the requirement for each of the two splice sites. This is currently missing.

Response 2: Thank you for raising this point. We have sequenced the intron lariat crossing the junction of branch point and 5' end of the intron (Figure 3e), as well as the mRNA across the splice sites (Figure S6d). Per reviewer's suggestion, we now conducted Nanopore sequencing of poly(A) selected RNA (**Figure S2** and response 1). We did not find transcripts contain 24×MBS in these RNA pool, consistent with the imaging results that the MBS sequences are located in the circular intron. The splice sites at the exon1a-exon2 junction are consistent in the two methods, and the same 5' sites were connected to the branch sites in the intron.

3. In the patient fibroblast analysis (Fig. 2), no validation was provided to ensure the specificity of either intron or exon probes.

Response 3: We thank the reviewer for raising this critical point. To validate the specificity of the intron and exon probes, we performed additional knock-down control experiments (**Figure S4a-c**). We used antisense oligonucleotide (ASO) to knock down *C9ORF72* expression in U-2 OS cells. Both intron and exon RNA level decreased by half in *C9ORF72*-ASO transfected cells compared to non-targeting control quantified by qRT-PCR (Figure S4c). We then performed two-color smFISH using the *C9ORF72* intron and exon probes at the same time. The average numbers of intron and exon per cell decreased to 50% upon ASO treatment (Figure S4a,b), consistent with the qRT-PCR result (Figure S4c). This demonstrates that the intron and exon probes used in this work are specific for the *C9ORF72* gene.

4. Two peculiarities were unexplained in Fig. 2. First, the intron probes did not detect any RNA foci. Second, nuclear intron signals were similar between control and C9 patient cells, which contradicted Fig. 1e.

Response 4: We thank the reviewer for raising this point. We think the gene expression level influences the nuclear granule visualization. The *C9ORF72* expression in patient fibroblast is very low. We chose fibroblast cell because these cells are well spread out such that the quantification of RNA spatial distribution (especially in cytoplasm) is accurate. We now included smFISH data from lymphoblast cells (**Figure S4d**). These cells have higher *C9ORF72* expression and therefore

have RNA foci in the nuclei of patient cells but not in control. This showed the intron probes can detect RNA foci.

5. Attempting to show that the intron puncta are in a lariat form, the authors treated fixed cells with RNase R, and found that the cytoplasmic intron puncta were somewhat resistant. Even for extracted RNAs, RNase R resistance could be caused by reasons other than circularity (e.g., PMID: 31269210). Here, the interpretation is further confounded by the presence of all of the proteins bound RNAs, which might also interfere with RNase R digestion. As a result, even the spliced exonic RNA was not completely removed (Fig. 3c). A definitive method to test circularity is by 2D denaturing gels, which have been used to isolate intron lariats (PMID: 27473169).

Response 5: We thank the reviewer to raise this important point about the RNase R treatment experiment. Indeed, if we use paraformaldehyde to fix the cell, RNase R is unable to digest the RNA, possibly due to the RNA binding proteins cross-linked with RNA. We used a special Methanol-Acetic Acid mixture to fix the sample to avoid cross-linking (See Methods). We agree with the reviewer that the RNA degradation efficiency by RNase R might be influenced by other factors. That is exactly why we performed a series of control experiments to show that the RNase R digestion is efficient when the same elements are located in linear RNAs under the same condition. We do not expect to have 100% elimination of linear RNAs, as this might indicate over-digestion. First, endogenous PolR2A mRNA and the exonic reporter mRNAs were efficiently degraded by RNase R (Figure S6a-c). This showed the RNase R coupled FISH experiment successfully degraded known linear RNAs, and the PBS structure and PP7 coat protein (PCP) do not affect the digestion. Second, the reporter mCherry-24×MBS were degraded efficiently by RNase R (Figure S5a,b), verifying that the MBS structure and MS2 coat protein (MBS) also do not inhibit the RNase R cleavage. Third, to further demonstrate that the complex structure introduced by GGGGCC repeats do not inhibit RNase R, we now performed additional control experiment. We constructed a cell line stably expressing an mRNA reporter with 70×GGGGCC repeats followed by 24×MBS (a normal capped linear transcript). We treated the cells with RNase R (Figure S5c-e) and the mRNAs were efficiently degraded. This demonstrated that a linear RNA containing both GGGGCC repeats and MBS sites can be efficiently degraded by RNase R treatment. Therefore, under the same condition, the resistance of (GGGGCC)₇₀-24×MBS intronic RNA to the RNase R treatment supports its circular form. We now emphasized these technical details in the main text and material and methods (Page 7 line 18 to Page 8 line 2, Page 23 lines 18-19).

We also agreed with the reviewer that a 2D denaturation gels may be a great way to visualize circular RNA fractions. However, this method requires large amount of RNA. In our system, there are only ~25 cytoplasmic intron molecules per cell. It is extremely challenging to detect low amount of RNAs using this technique. We therefore rely on RNase R treatment, which is a standard procedure for evaluation of circular RNA forms with proper controls.

6. It was not explained why the stabilized, nuclear introns were efficiently digested by RNase R (Fig. 3d).

Response 6: We now included a sentence to explain this point (Page 8 lines 2-6). Nuclear introns may exist in two forms: unspliced pre-mRNA and spliced intron. In the RNase R assay, we did not separate signals into two individual groups. The pre-mRNA is linear and can be digested by RNase R, contributing to the reduction shown in the nucleus. In addition, a portion of the spliced intron may still be debranched to linear form and sensitive to RNase R treatment. However, some nuclear introns maintain the circular form and are exported to the cytoplasm, where they accumulate over time.

7. The translation analysis in Fig. 5 was not sufficiently explained. What percent of cytoplasmic intron RNAs are translated? Fig. 5e shows that multiple nascent chains were found on each RNA, which is surprising considering that RAN translation is known to be inefficient.

Response 7: We thank the reviewer for pointing it out. The reviewer is right that RAN translation is very inefficient compare with canonical ATG translation, which agrees with our data. We now explained the translation efficiency more clearly in the manuscript (Page 10 lines 18-21). In an ensemble experiment, the protein production rate depends on two factors: the fraction of translating mRNAs and the number of ribosomes on the translating ones. The single molecule approach provides more details. First, very low fraction of RNA with repeat expansion undergoes RAN translation: 3% compare to 50-90% for canonical AUG translation. Second, the number of nascent peptides on RAN translating RNA is also less: mostly 1-2 in RAN translation, compared with 5 to 10 in canonical translation (with similar design). So it appears that most repeat RNA are not undergoing RAN translation. But when they are translating, they can be translated by more than one ribosome.

8. Perhaps the most provocative claim is that the capless intron lariats can be translated, which would require an IRES-like initiation mechanism. The authors should test the C9 intron sequence for IRES activity with *in vitro* transcribed RNAs to avoid common caveats of cellular IRES assays (PMID: 20576611).

Response 8: We appreciate the reviewer's recognition of the significance of our model and suggestion of the *in vitro* experiment. To further demonstrate the "IRES-like" cap-independent initiation mechanism, we performed *in vitro translation* assay. We generated *in vitro* transcribed Nanoluc luciferase (NLuc) RNAs without caps and performed translation assay in rabbit reticulocyte lysate. The RNAs containing repeats in front of NLuc (without ATG) in both GA and GP reading frames produced substantially higher luciferase activity compared with negative control without repeats (**Figure S10b**). This data is in line with our previous finding that GGGGCC repeats containing C9 intron sequence can be translated without 5' cap (PMID: 29302060).

9. A previous study from the same group (PMID: 31587919) has shown that NXF1 and NXT1 also promote the nuclear export of linear GGGGCC repeat RNAs. Therefore, it appears to be the repeat sequence, rather than circularity that determines its export pathway.

Response 9: We thank the reviewer for raising this point. We agree with the reviewer and this is what we intend to propose that the repeat expansion mediates this process. NXF1 / NXT1 export pathway is a universal export pathway. It is possible that some RNA binding proteins (RBPs) bind GGGGCC repeats and mediate this process. Therefore, we believe that circularization stabilize the

intron and RNA binding proteins on the repeats mediate the RNA export through the NXF1 /NXT1 pathway. We now included it in the discussion (Page 15 lines 15-16).

Minor issues

1. Results in Fig. 3e is expected by design. They do not show that intron puncta are predominantly lariats.

Response 10: We agree with the reviewer that Fig. 3e only identified the sequence of the intron lariat and did not directly demonstrate the intron spots in the images are predominantly lariats. It is technically challenging to directly show lariat in the puncta. Multiple experiments, including sequencing, imaging and RNase R treatment, altogether suggest this possibility by reasoning. We changed the text to reflect this (Page 8 lines 8-9).

2. The authors found that intron stabilization and export was specific to G-rich repeats including GGGGCC and CGG, but not to other disease-associated repeats. However, CCTG repeats in DM2 are also intronic and RAN-translated. The authors should discuss the apparently distinct RAN translation mechanisms between these two repeat expansion mutations.

Response 11: We thank the reviewer for raising this point. RAN translation products of CCTG repeats expansion have been found in DM2 patients (PMID: 28910618). We would like to emphasize that all the previous studies used reporter construct of pure CCTG repeats in mRNA format, not in the intron of *ZNF9* gene. Our work is the first attempt to study CCUG RAN translation in the splicing context. It is possible that other intronic elements in the host gene can promote nuclear export, but were not included in our reporter (the endogenous intron is very long. Only the regions close to the two splice sites were kept in the reporter). It is also likely that there is slow production of poly-peptides from the low level of cytosolic intron RNA. Therefore, we did not propose there are distinct RAN translation mechanisms, but the relative efficiency could be different. The amount of the accumulated poly-peptides could be influenced by the different molecular properties of the repeat RNA templates. This might give hints to the distinct pathogenic mechanisms in different diseases. For example, the RNA granule sequestration of RBPs might be the predominant mechanism in some repeat expansion diseases (such as DM2), while the RAN translation product could be the driving toxicity in some other repeat expansion diseases. We now included a few sentences in the discussion (Page 14 lines 3-7).

3. A typo (“Raipd”) in Fig. 6e.

Response 12: Thank you! We now corrected it.

4. A space is missing in Page 3 Line 22.

Response 13: Thank you! We now corrected it.

Reviewer #2

In this excellent study by Wang et al, the authors use advanced imaging methods to investigate the localization and translation of the circular repeat-containing intronic RNA C9ORF72. Using a clever reporter construct incorporating the intron and neighboring exons, they exploit the MS2/PP7-RNA reporter system as well as the SunTag-reporter system to investigate the spatiotemporal regulation of repeat RNA localization and translation. They find the hexanucleotide repeat expansion to be within a circularized intron, that is exported from the nucleus and translated. Validation of these findings in C9ORF72-ALS patient samples and analysis of the export pathway is further strengthening the paper. The paper is very well written and the data are well presented making it easy to follow the story. Overall, the results are novel, original, very convincing and the paper is a significant and major advance for the field. Also, I would like to stress that the results have a significant and high impact for a large and interdisciplinary community justifying publication with high priority.

We appreciate the reviewer's positive evaluation for our manuscript.

1. There is a couple of points the authors could consider to improve their story: Whereas the paper is well written, the figure legends would certainly benefit from another editing attempt. Clearly, better explanation in figure legends / annotations within figures would further improve readability and do more justice to the impressive data.

Response 1: Thank you! Per the reviewer's suggestion, we edited the figure legend to include more details to improve the readability.

2. In addition, the SunTag portion could be improved (Fig. 5). While the data from the SunTag experiment look very convincing, a control including a translation inhibitor is crucial in my opinion to exclude possible artefacts due to aggregation of SunTag-reporters without active translation. Possibly, the authors have done this already, but I suggest incorporating this data into the main figure.

Response 2: We thank the reviewer for the suggestion. We now included the control experiment using translation inhibition puromycin (**Figure 5b**). As expected, puromycin rapidly removed nascent peptides from the intron RNAs.

3. Finally, Fig. 1 could be improved. For improved understanding of Fig. 1b, it would help to include a depiction of the smFISH probes and possibly exclude the MS2 coat protein and PP7 coat protein labelling, since the analysis in Fig. 1 was performed solely using smFISH. ‘

Response 3: Thanks for the good point. We now modified Figure 1 and the legend as suggested by the reviewer.

4. Furthermore, the y-axis title in Fig. 1c is misleading, I guess displayed are the number of nuclear RNA granules normalized to the total number of RNA foci per cell.

Response 4: Fig. 1c shows the number of nuclear RNA granules per cell. Not all cells have nuclear RNA granules. That is why the average number is less than 1.

Reviewer #3

C9ORF72 hexanucleotide GGGGCC repeat expansion is the most common cause of ALS and FTD. Repeat-containing RNA mediates toxicity through nuclear granules and dipeptide repeat (DPR) proteins produced by repeat-associated non-AUG translation. In this manuscript, the authors present the data suggesting that the splicing intron with G4C2 repeats is stabilized in a circular form, which could serve as the translation template. They further showed that the NXF1-NXT1 pathway plays an important role in the nuclear export of the circular intron and modulates toxic DPR production. Although the results are potentially interesting, however, all the analyses were performed in vitro and the main conclusion was based on in vitro overexpression system. Additional in vivo and functional data are needed to support the authors' conclusion. Below are some specific comments:

1. The authors did demonstrate the presence of circular intron in cytoplasm, however, it is unclear whether linear intron containing G4C2 repeat is present in cytoplasm. If it does, what's the ratio between linear and circular forms, particularly the transcripts produced from endogenous locus? The linear intron RNA could potentially be present in granules, which would make it difficult to detect using the techniques employed in this manuscript.

Response 1: We appreciate the reviewer raised this point. The reviewer is right that the RNA in the nuclear granule might be linear, because they are sensitive to RNase R treatment, as quantified in Figure S6c. We now clarified this point in the text (Page 8 lines 2-6). When possible, we incorporated the data from endogenous locus to validate the results from our reporters. We showed that circular intron exists in the cytoplasm of C9-ALS patient fibroblast; repeat containing introns form nuclear RNA granules in patient lymphoblast; the DPR is reduced when we knocked down NXF1/NXT1. Like any studies, our techniques have their limitations. We cannot accurately quantify the ratio between linear and circular forms. But we believe they also have advantages over other ensemble biochemistry methods. It provides spatial distribution of different RNA species and evidence that circular introns are the substrate for RAN translation, which provides significant novel knowledge to better understand the disease mechanism.

2. Instead of using patient fibroblast cells, a disease-relevant cell type(s) should be examined instead.

Response 2: We appreciate the reviewer's suggestion. With patient derived fibroblast cells, we have demonstrated that the spliced intron instead of pre-mRNA gets exported. This is interesting for both biology and disease. We chose fibroblast cell because these cells are well spread out such that the quantification of RNA spatial distribution in cytoplasm by imaging between control and patient cells is accurate. It has been shown that there are RAN translation products in multiple cell types, suggesting the basic molecular pathways are conserved among cell types. Therefore, we think it is not unreasonable to use patient fibroblast cells to study the basic molecular mechanism of repeat RNA processing. We agree with the reviewer that a disease-relevant cell type would be even better, such as neurons. However, it is technically challenging to perform the single molecule FISH experiment in the neuron culture due to high background signals. Furthermore, the neurons have small cell body and highly complex neurites. As C9ORF72 is among the low expressed genes

in transcriptome (higher in neurons but still a low expressed gene), it is extremely difficult to quantify the cytosolic molecules accurately. We hope to optimize the technique to be able to study the regulator of the molecular pathways in neurons in future.

3. The authors suggested that they tested expanded FMR1 CGG repeat. The size of CGG repeat that they used is 34, which is the normal allele of FMR1.

Response 3: We thank the reviewer for raising this point. We agree longer repeats will be more disease relevant. We previously had difficulty to make longer CGG repeats stable in our reporter construct. Now we successfully cloned a (CGG)₉₈ repeat in the reporter and found similar results. All of the data previously obtained with the (CGG)₃₄ reporter were replaced with the new (CGG)₉₈ reporter (**Figure 3, S8 and S9**).

4. The results of NXF1-NXT1 pathway analysis are interesting, however, it was only done in vitro. Could NXF1-NXT1 pathway could modulate the G4C2 repeat toxicity in vivo?

Response 4: We thank the reviewer for the suggestion. We have confirmed that the NXF1-NXT1 pathway modulates the endogenous DPR levels in C9-ALS patient derived induced pluripotent stem cell (iPSC) differentiated neurons (iPSN) (Figure 6d). Testing the function of NXF1-NXT1 in regulation of the G4C2 repeat induced toxicity in mouse model and development of therapeutic strategy would be important future research, but we think it is beyond the scope of the current manuscript.

Reviewers' comments:

Reviewer #1 (Remarks to the Author):

In the revised version of their manuscript, Wang et al. addressed some of my original comments but not all of them, as detailed below. Most importantly, as also pointed out by another reviewer, although the authors showed that circular introns indeed existed in cytoplasm, they could not rule out the possibility that the translated RNA represented a small population of linear RNAs. Therefore, the main conclusion of the paper that circular intron RNAs are translated remains unjustified.

Original major concern #1: The authors performed Nanopore-based RNA sequencing to "validate" the use of this reporter construct. However, out of a mere 171 reads that can be mapped to the reporter, only 24 (14%) reads contained the expected exon-exon junction. Fig S2 shows at least 6 reads that were spliced in unintended ways, and 46 reads (Group 3+4) that have unintended 3' ends. In other words, this construct expressed more erroneous RNAs than correct ones. Because the alleged translated introns are relatively rare events (3% of all intron RNAs), they can easily be artifactual products expressed from this problematic construct (e.g., additional Group 4 transcripts with detectable MBS repeats) that has not been captured by the analysis of only 171 reads.

Original major concern #2: Unfortunately, the authors misunderstood the comment and thus failed to address it. The critical control experiment I requested was that when either of the two splice sites was disrupted, the authors should no longer observe cytoplasmic MBS-only puncta.

Original major concern #3: For unknown reason, the authors chose to confirm the specificity of their smFISH probes in U2OS cells, but not in patient fibroblasts. This is problematic because the observe cytoplasmic "intron" puncta in fibroblasts can still be from nonspecific targets that are not present in U2OS cells. These knockdown experiments should have been done in fibroblasts, so that the authors could see whether ASO can decrease both exon and intron smFISH signals.

Original major concern #8: The authors performed a cursory analysis and claimed that G4C2 repeats have IRES activity. However, these in vitro translation assays are known to have reduced requirement for 5'-cap structure. The large difference observed between their negative control and 70xG4C2 luciferase reporters could be due to the higher stability of the 70xG4C2 RNA in the presence of 5'-3' exonuclease activity. To increase the specificity of the assay, the authors should use bicistronic mRNAs with GC content- and length-matched inserts (e.g., Firefly luciferase (Fluc)-70xG4C2-Nluc versus Fluc-70xG2C4-Nluc). In addition, positive controls such as viral IRESs should be included.

Reviewer #2 (Remarks to the Author):

The authors addressed all of my comments in a very concise and productive manner, I do not have any concerns left. I was particularly impressed by the authors's responses to the concerns of Ref. 1. I have rarely seen more impressive and positive responses than those deposited. Overall, it is a great study that has been improved substantially by the authors. Congratulations! I full heartedly recommend publication with very high priority.

Reviewer #3 (Remarks to the Author):

In the revised version, the authors answered all the questions raised by the reviewers. I appreciate their effort to address my concern; however, I think it is still premature to be accepted for the publication due to the following reasons:

1. In the revised manuscript, the authors present their Nanopore long-read direct RNA sequencing and conclude that the predominant MS2 signal in the smFISH experiment represents the spliced intron, not due to misprocessed mRNA. However, the sequencing depth of the presented Nanopore RNA sequencing is very low, 1.53 million raw seq with 24 sequences mapped to the reporter construct. Additional sequencing is needed to support the authors' conclusion.
2. The claim that the excised intron lariats are stabilized, exported, and translated is still only supported by in vitro reporter cell lines and patient cell lines. There is neither in vivo data nor the demonstration of functional relevance to ALS pathogenesis/FXTAS.

Response to reviewers' comments

Reviewer #1 (Remarks to the Author):

In the revised version of their manuscript, Wang et al. addressed some of my original comments but not all of them, as detailed below. Most importantly, as also pointed out by another reviewer, although the authors showed that circular introns indeed existed in cytoplasm, they could not rule out the possibility that the translated RNA represented a small population of linear RNAs. Therefore, the main conclusion of the paper that circular intron RNAs are translated remains unjustified.

Response: There are several findings from our paper. Translation of circular intron is one of them. First, we showed that the repeat expansion can stabilize the spliced intron in a circular form. Second, the repeat expansion mediates the nuclear export of the circular intron. Third, the circular intron RNA in the cytoplasm can serve as the template for RAN translation. Fourth, the NXF1-NXT1 pathway mediates the repeat-containing circular intron export, thereafter regulates the dipeptide production in patient cells. Fifth, the G-rich repeats are important for intron stabilization and export. We agreed with the reviewer that there is always a small chance that a small population of linear intron escapes nucleus and is translated. However, various evidence from the current feasible techniques all showed consistent results to support the conclusion that most of the repeat RNA in the cytoplasm is circular intron, and the circular intron RNA can be translated. We will tune down our claim in the revised manuscript: that the evidence is consistent with circular introns being translated. We hope the reviewer will recognize the multiple novel findings from our work.

Original major concern #1: The authors performed Nanopore-based RNA sequencing to “validate” the use of this reporter construct. However, out of a mere 171 reads that can be mapped to the reporter, only 24 (14%) reads contained the expected exon-exon junction. Fig S2 shows at least 6 reads that were spliced in unintended ways, and 46 reads (Group 3+4) that have unintended 3' ends. In other words, this construct expressed more erroneous RNAs than correct ones. Because the alleged translated introns are relatively rare events (3% of all intron RNAs), they can easily be artifactual products expressed from this problematic construct (e.g., additional Group 4 transcripts with detectable MBS repeats) that has not been captured by the analysis of only 171 reads.

Response: In the original review, we were asked whether there are misprocessed transcripts that would result in mRNAs with MBS only, but without PBS, a potential explanation for smFISH images of cytoplasmic MBS signal. Because Next-gen Illumina sequencing would not be able to answer this question, we performed Nanopore sequencing on poly(A) selected RNA molecules directly. This technology has a larger error rate and lower coverage depth. We mapped 171 molecules out of 1.52 million total reads to our reporter constructs, a typical performance for this technology (PMID: 31931956 and 31740818). To further increase the coverage depth, we transiently transfected the splicing reporter into 293T cells and sequenced the poly(A) RNA (Fig. S3). As expected, we obtained **40 folds more relevant reads**: from the 0.84 million raw sequences, we identified 6838 reads that can be aligned to the reporter region with high confidence. Among these reads, **865 mRNA sequences** span from the 5' (Exon1a) to the expected poly(A) sites (Group 1). The exon-exon junction site was the same as our stable line and agreed with the isoforms we identified from the intron and mRNA amplicon sequencing experiment (Figure S7d). There were

only 3 reads (Group 4) with a small portion of MBS region that also contain PBS, which is 0.04% of the total reads. **There were no MBS-only sequences.** This further demonstrated that the Group 4 reads identified from our stable cell line may come from recombination errors during the genomic integration which only happens in rare cases. With these further sequencing experiment, we believe the **evidence we have achieved is sufficient to support our imaging experiment and conclusion for the following reasons:**

- 1) The majority of transcripts (both group 1 and group 2) are consistent with the major isoforms that we studied. Not all the reads can be sequenced through the 24x PBS to the splicing junction site because of the premature sequencing termination, a limitation of the technique. However, the main point is that all the reads (~900 total) that were sequenced till the 5' end showed correct splicing. We did not identify any mis-spliced transcripts. We also did not find unspliced transcripts. This showed that the reporter transcript has efficient and correct splicing, and could support our interpretation of the imaging data.
- 2) Out of the total 2.36 million reads, we did not find any RNA molecules containing 24xMBS by mapping to the MBS fragment only. If the MBS+ only RNA is from mis-spliced transcript but not spliced intron, we should be able to easily find reads mapping to MBS sequences as these transcripts contain poly(A) tail. However, we absolutely did not find such reads. This data has clearly excluded the possibility of MBS-only poly(A) mRNA produced from the reporter: therefore, the MBS-only puncta in the cytoplasm is most likely intron, instead of poly-A mRNA.
- 3) The reads in group 3 have different 3' end before the repeat expansion and MBS sites. The isoforms and short transcripts are often poorly annotated in the genome, especially for low-expressed genes, such as *C9ORF72*. In fact, we analyzed the published dataset of Nanopore RNA-seq (PMID: 31740818). There were 28 reads mapped to the endogenous *C9ORF72* gene, and 5 of them had 3' end in various positions in introns. Therefore, we cannot exclude the possibility that these are real poly(A) sites (if not technical error) used by the endogenous gene instead of “erroneous RNAs” from the reporter. Hope you can appreciate how difficult it is to sequence such low abundant genes and agree with us that this does not necessarily mean “this construct expressed more erroneous RNAs than correct ones”. Nevertheless, the potential alternative poly(A) sites in group 3 are not relevant to the imaging study of repeat RNA at all, as these transcripts contain neither repeat expansion nor 24xMBS sites. These molecules are not visible and will not affect the explanation of the imaging experiment.
- 4) The group 4 reads are most likely due to recombination errors during the genomic integration for stable cell line. Out of the 1.52 million reads from stable U-2 OS cells, we identified 5 molecules that only partially map to the transgene, with the remaining fragments containing random sequences. We apologize this was not clearly annotated in the Figure S2, as we were only able to show the regions that can be mapped to the transgene but failed to indicate this was not the full length of the reads. We think this is most likely due to recombination errors during the genomic integration step. In the transient transfection experiment, we did not observe such events. This was not caused by incorrect splicing of the transgene. Furthermore, these reads contained less than 12 copies of MBS, which is invisible for single molecule experiment and will not influence our main

conclusion. This type of error should only be visible in a very small subset of cells if any rather than the whole population. As we always quantified hundreds of cells in the imaging experiments, although only 3% introns have been translated, we are still very confident that they are translated from spliced intron.

In the end we would like to underline that we analyzed total 2.36 million raw reads, and 7009 reads align to our reporter (mapped to either MBS or PBS). This should have ensured that all of the MBS poly(A) RNA derived from problematic misprocessing would have been identified if there were any. We did NOT find any isoforms containing 24xMBS, showing that the MBS signal in single molecule imaging cannot be from the misprocessed mRNA.

Original major concern #2: Unfortunately, the authors misunderstood the comment and thus failed to address it. The critical control experiment I requested was that when either of the two splice sites was disrupted, the authors should no longer observe cytoplasmic MBS-only puncta.

Response: We apologize for misunderstanding the reviewer's intention. We built a reporter in which all the potential 5' splice sites in the exon1 were removed and performed smFISH in the stable cell line. More than 75% of the cytoplasmic MBS signal is colocalized with PBS for the mutant, compared with 1.4% for the wide type splicing reporter (Figure S4f). The colocalization not being 100% could be explained by imperfect detection. For example, a 90% detection efficiency in both colors would result in 81% colocalization. This demonstrates that the cytoplasmic MBS alone puncta in the wide type reporter depend on the presence of splicing site.

Original major concern #3: For unknown reason, the authors chose to confirm the specificity of their smFISH probes in U2OS cells, but not in patient fibroblasts. This is problematic because the observe cytoplasmic "intron" puncta in fibroblasts can still be from nonspecific targets that are not present in U2OS cells. These knockdown experiments should have been done in fibroblasts, so that the authors could see whether ASO can decrease both exon and intron smFISH signals.

Response: We chose the human cell line U-2 OS to validate the FISH probes for *C9ORF72* as its transcript level is higher than fibroblast and we felt it is more convincing to show a reduction. Nevertheless, we now performed the knockdown experiment in patient derived fibroblast cells, as requested by the reviewer. There was 3 times knockdown of both intron and mRNA quantified by qRT-PCR. The average numbers of intron and exon per cell from the smFISH experiment decreased to 37% upon ASO treatment (Figure S4a, b), consistent with the qRT-PCR result (Figure S5e-g). This demonstrates that the intron and exon probes used in this work are specific to the *C9ORF72* gene in patient fibroblast cells.

Original major concern #8: The authors performed a cursory analysis and claimed that G4C2 repeats have IRES activity. However, these in vitro translation assays are known to have reduced requirement for 5'-cap structure. The large difference observed between their negative control and 70xG4C2 luciferase reporters could be due to the higher stability of the 70xG4C2 RNA in the presence of 5'-3' exonuclease activity. To increase the specificity of the assay, the authors should use bicistronic mRNAs with GC content- and length-matched inserts (e.g., Firefly luciferase (Fluc)-70xG4C2-Nluc versus Fluc-70xG2C4-Nluc). In addition, positive controls such as viral IRESs should be included.

Response: We apologize that the in vitro translation experiment we have performed was not what was asked. We now performed the assay using the bicistronic RNAs, by placing (GGGGCC)₇₀-NLuc after the stop codon of an open reading frame (Fig. S11b), as suggested by the reviewer. As positive controls, we used viral IRES to initiate translation of NLuc. However, we do not think the proposed negative control is appropriate. It is known that the RAN translation occurs in many expanded repeats, including both GGGGCC repeats and CCCC GG repeats. Similarly, secondary RNA structures can facilitate the initiation of non-canonical translation. Sequences with GC-content matched to GGGGCC may form unintended secondary structure. Therefore, we believe that the proper negative control should be length-matched but randomized sequence. We performed the in vitro translation assay and found that GGGGCC in both GA and GP frames induces much higher NLuc activities than negative control: the GA frame is even comparable to EMCV IRES (Supplementary Fig. 11c). This data support that GGGGCC repeats containing C9 intron sequence can be translated in cap-independent fashion.

Reviewer #2 (Remarks to the Author):

The authors addressed all of my comments in a very concise and productive manner, I do not have any concerns left. I was particularly impressed by the authors's responses to the concerns of Ref. 1. I have rarely seen more impressive and positive responses than those deposited. Overall, it is a great study that has been improved substantially by the authors. Congratulations! I full heartedly recommend publication with very high priority.

Response: We thank the reviewer for his/her comments and support of our work.

Reviewer #3 (Remarks to the Author):

In the revised version, the authors answered all the questions raised by the reviewers. I appreciate their effort to address my concern; however, I think it is still premature to be accepted for the publication due to the following reasons:

1. In the revised manuscript, the authors present their Nanopore long-read direct RNA sequencing and conclude that the predominant MBS signal in the smFISH experiment represents the spliced intron, not due to misprocessed mRNA. However, the sequencing depth of the presented Nanopore RNA sequencing is very low, 1.53 million raw seq with 24 sequences mapped to the reporter construct. Additional sequencing is needed to support the authors' conclusion.

Response: Because Next-gen Illumina sequencing (which has much high coverage depth) would not be able to answer this question, we performed Nanopore sequencing on poly(A) selected RNA molecules directly. This is a newly developed technique and still has a lot of limitations. This technology has a larger error rate, lower coverage depth and early truncation. To the best of our knowledge, the nanopore long reads sequencing is currently the only available methods to directly sequence long RNA. To increase the coverage, we now performed additional Nanopore sequencing using transient transfection. We were able to increase the coverage depth by 40 folds, and reached the same conclusion: there is literally no MBS-only polyadenylated RNA sequence. Please see the detailed response to the reviewer one's major concern #1.

2. The claim that the excised intron lariats are stabilized, exported, and translated is still only supported by in vitro reporter cell lines and patient cell lines. There is neither in vivo data nor the demonstration of functional relevance to ALS pathogenesis/FXTAS.

Response: In this particular study, we applied the cutting-edge single molecule imaging technique to study the potential disease mechanisms. Our findings provide a new angle to understand the disease. For the first time, we showed that the repeat expansion can stabilize the spliced intron in the circular form and mediate the intron export to the cytoplasm, and G-rich sequence is important for this phenomenon. Through live cell imaging, we showed RAN translation on circular intron, and the translation is rapidly up-regulated by stress. We also have demonstrated that the repeat-containing circular intronic RNA is exported to cytoplasm through the NXF1-NXT1 pathway and confirmed its function in patient iPS-neurons (in vivo data).

These novel findings are highly significant in understanding the disease mechanisms of repeat expansion diseases, as previous studies failed to recognize the importance of the location of repeat expansion and ignored the influence by splicing. As there have been extensive studies demonstrating the toxicity of poly-dipeptides produced from the RAN translation, understanding the molecular mechanisms how the dipeptides are produced from an intronic repeat is directly relevant to ALS/FTD pathogenesis.

Many high-impact publications on the basic mechanisms of dipeptide toxicity have used in vitro assays or transgene expression cell lines, without any experiments in patient cells or animal models (such as PMID: 25081482, 26406374, 28306503). The functional relevance of basic molecular mechanism is well appreciated and has been shown to give great impact on understanding disease pathogenesis in the following studies in vivo. We think our study has identified several novel molecular properties and pathways of the repeat RNA metabolisms that have direct relevance to the dipeptide production. This paves the path and shows the direction for further studies.

REVIEWERS' COMMENTS

Reviewer #1 (Remarks to the Author):

I appreciate the authors for their effort to address the concerns. Their additional sequencing experiment yielded more convincing results supporting the validity of their splicing reporter. However, the authors mentioned in their response that in the "the published dataset of Nanopore RNA-seq (PMID: 31740818), there were 28 reads mapped to the endogenous C9ORF72 gene, and 5 of them had 3' end in various positions in introns." Wouldn't this suggest that endogenous C9orf72 repeats may be present as prematurely polyadenylated RNA, a possibility that has not been considered in this study?

Furthermore, the critical concern of experimental artefacts remains with the translation reporter, which the authors have not performed any sequencing validation on. This reporter contains additional Suntag/AID sequences and is likely even more prone to cryptic splicing. Without such validation, the extraordinary claim that intron lariats are translated remains unjustified.

Reviewer #3 (Remarks to the Author):

The increased sequencing depth has provided additional support to the authors' conclusion. Although I still have concerns with all in vitro data, however, I would support the publication in Nature Communications.

Reviewer #4 (Remarks to the Author):

RAN translation of dipeptide repeats (DPRs) is templated by sense (G4C2) and antisense (G2C4) repeat expansions in the first intron of C9orf72. Due to the intronic location of this expansion, several C9 RNAs have been proposed to mediate DPR translation, including both partially spliced linear C9 RNA and some form of circRNA. Here, Wang and colleagues use single molecule imaging to provide evidence for the latter mechanism, specifically an intron 1 circular intronic (ci)RNA that is resistant to debranching and stable. The stability and export of C9 ciRNA is dependent on the G4C2 repeat, or just a GC-rich repeat since CGG repeats also promote export, and export requires NXF1-NXT1 activity. Finally, they provide evidence that C9 ciRNA template RAN translation. Overall, the C9 ciRNA stability and export data are convincing, and while the evidence for DPR translation from patient C9orf72 G4C2 endogenous mutant allele is weaker. However, the authors have responded adequately to the remaining concerns of the previous reviewers, so I have only minor points to address.

1. Abstract. The sentence "C9ORF72 hexanucleotide GGGGCC repeat expansion is the most common cause of amyotrophic lateral sclerosis (ALS) ..." should be modified to "C9ORF72 hexanucleotide GGGGCC repeat expansion is the most common genetic cause..."

1. Fig. 1a. The intron illustration should be improved to show the relative positions of the 24xMBS and the G4C2 repeat within a conventional intron lariat.

2. Fig. S2 and S3. For Fig. S1 Group 1, it appears from the nanopore reads that only 1 read mapped to the correct 5' splice site with most junctions mapping more 5' within e1a – is this correct and what is the explanation for this upstream 5' splice site? For S3, Group 1 and Group 2 show 3' extensions of exon 1 into the intron – please address.

3. Fig. 4 legend. The figure indicates (CGG)₃₄ but the legend (b,c) states (CGG)₉₈.

4. Discussion, p.13. The authors mention that the GGGGCCexp 'modestly inhibits splicing' but their results were obtained using a relatively short (G4C2)₇₀ construct – would this still be the case for the much larger expansions typical of C9-ALS/FTD?

Response to reviewers' comments

Reviewer #1 (Remarks to the Author):

I appreciate the authors for their effort to address the concerns. Their additional sequencing experiment yielded more convincing results supporting the validity of their splicing reporter. However, the authors mentioned in their response that in the "the published dataset of Nanopore RNA-seq (PMID: 31740818), there were 28 reads mapped to the endogenous C9ORF72 gene, and 5 of them had 3' end in various positions in introns." Wouldn't this suggest that endogenous C9orf72 repeats may be present as prematurely polyadenylated RNA, a possibility that has not been considered in this study?

Furthermore, the critical concern of experimental artefacts remains with the translation reporter, which the authors have not performed any sequencing validation on. This reporter contains additional Suntag/AID sequences and is likely even more prone to cryptic splicing. Without such validation, the extraordinary claim that intron lariats are translated remains unjustified.

Response: We thank the reviewer for the recognition of our effort. For the published Nanopore RNA-seq of endogenous C9ORF72 transcript, the data is from control samples without repeat expansion. And the alternative 3' end of all the 5 reads are not in the downstream intron region of the GGGGCC repeats. Therefore, even these are real prematurely polyadenylated RNA, they will not contain the repeats.

The translation reporter has exactly the same gene structure as the splicing reporter. We have used RT-PCR to confirm that the transcripts were efficiently spliced in both reporters. The Nanopore long reads sequencing showed consistent results of the splicing junctions as the PCR-based method in the splicing reporter. As the translation construct has the same splicing product and efficiency, we think the splicing of the translation reporter is consistent to the splicing reporter.

Reviewer #3 (Remarks to the Author):

The increased sequencing depth has provided additional support to the authors' conclusion. Although I still have concerns with all in vitro data, however, I would support the publication in Nature Communications.

Response: We thank the reviewer for his/her comments and support of our work.

Reviewer #4 (Remarks to the Author):

RAN translation of dipeptide repeats (DPRs) is templated by sense (G4C2) and antisense (G2C4) repeat expansions in the first intron of C9orf72. Due to the intronic location of this expansion, several C9 RNAs have been proposed to mediate DPR translation, including both partially spliced linear C9 RNA and some form of circRNA. Here, Wang and colleagues use single molecule imaging to provide evidence for the latter mechanism, specifically an intron 1 circular intronic (ci)RNA that is resistant to debranching and stable. The stability and export of C9 ciRNA is

dependent on the G4C2 repeat, or just a GC-rich repeat since CGG repeats also promote export, and export requires NXF1-NXT1 activity. Finally, they provide evidence that C9 ciRNA template RAN translation. Overall, the C9 ciRNA stability and export data are convincing, and while the evidence for DPR translation from patient C9orf72 G4C2 endogenous mutant allele is weaker. However, the authors have responded adequately to the remaining concerns of the previous reviewers, so I have only minor points to address.

Response: We thank the reviewer for the support of our work.

1. Abstract. The sentence “C9ORF72 hexanucleotide GGGGCC repeat expansion is the most common cause of amyotrophic lateral sclerosis (ALS) ...” should be modified to “C9ORF72 hexanucleotide GGGGCC repeat expansion is the most common genetic cause...”

Response: Thank you! We now corrected it.

1. Fig. 1a. The intron illustration should be improved to show the relative positions of the 24xMBS and the G4C2 repeat within a conventional intron lariat.

Response: Thank you! We now modified the diagram in Figure 1a and several other figures as suggested by the reviewer.

2. Fig. S2 and S3. For Fig. S1 Group 1, it appears from the nanopore reads that only 1 read mapped to the correct 5'ss with most junctions mapping more 5' within e1a – is this correct and what is the explanation for this upstream 5'ss? For S3, Group 1 and Group 2 show 3' extensions of exon 1 into the intron – please address.

Response: Thank you for raising this good point. Actually, there are three alternative 5' splice sites in the exon1 (See FigS7d). The one “correct” exon in FigS2 is the isoform-2 (NM_1450056.6). The most abundant mRNA isoform we found is the one that uses the 5'SS-1 site (DB079375.1). The results in Fig S2 are consistent with this finding. In FigS3, as the sequencing depth is increased, we have the chance to identify the isoform-3 (NM_001256054.2), which is the rarest one and longest isoform. In conclusion, the reads with three different boundaries in exon1 represent three different alternative 5' splice sites, which are also found in the endogenous *C9ORF72* transcripts. We now edited the diagram in FigS2 and S3 to annotate the three isoforms more clearly.

3. Fig. 4 legend. The figure indicates (CGG)₃₄ but the legend (b,c) states (CGG)₉₈.

Response: Thank you! We now corrected it.

4. Discussion, p.13. The authors mention that the GGGGCCexp ‘modestly inhibits splicing’ but their results were obtained using a relatively short (G4C2)₇₀ construct – would this still be the case for the much larger expansions typical of C9-ALS/FTD?

Response: Thank you for pointing out this question. We mentioned that the GGGGCCexp ‘modestly inhibits splicing’ based on the smFISH results of endogenous *C9ORF72* transcripts in patient cells (Fig2C), which have >500 repeats. As we showed that the slightly increased unspliced pre-mRNA retained in nucleus and was not exported to cytoplasm, we believe the spliced intron serves as the predominant template for RAN translation.